# Spatial variability of heavy metal concentration in urban pavement joints – A case study

Collin J. Weber[1*], Alexander Santowski[1], Peter Chifflard[1]

[1]Department of Geography, Philipps-University Marburg, Marburg, 35037, Germany

*Correspondence to*: Collin J. Weber (collin.weber@geo.uni-marburg.de)

**Abstract**

Heavy metals are known to be among one of the major environmental pollutants especially in urban areas and, as generally known, can pose environmental risks as well as direct risks to humans. This study deals with the spatial distribution of heavy metals in different pavement joints in the inner-city area of Marburg (Hesse, Germany). Pavement joints, defined as the joint

between paving stones and filled with different materials, have so far hardly been considered as anthropogenic materials and potential pollution sources in urban areas. Nevertheless, they have an important role as possible sites of infiltration for surface runoff accumulation areas, and are therefore a key feature of urban water regimes. In order to investigate the spatial variability of heavy metals in pavement joints, a geospatial sampling approach was carried out on six inner-city sampling sites, followed by heavy metals analyses via ICP-MS, and additional pH and organic matter analyses. A first risk assessment of heavy metal

pollution from pavement joints was performed.

Pavement joints examined consist mainly of basaltic gravel, sands, organic material and anthropogenic artefacts (e.g., glass, plastics) with an average joint size of 0.89 cm and a vertical depth of 2 – 10 cm. In general, the pavement joint material shows high organic matter loads (average 11.0% by mass) and neutral to alkaline pH values. Besides high Al and Fe content, the heavy metals Cr, Ni, Cd and Pb are mainly responsible for the contamination of pavement joints. The identified spatial pattern

of maximum heavy metal loads in pavement joints, could not be attributed solely to traffic emissions, as commonly reported for urban areas. Higher concentrations were detected at runoff accumulation areas (e.g., drainage gutters), and at the lowest sampling points with high drainage accumulation tendencies. Additional Spearman correlation analyses show clear positive correlation between runoff accumulation value and calculated exposure factor (ExF) ($r_{sp} = 0.80$; $p < 0.00$). Further correlation analyses revealed different accumulation and mobility tendencies of heavy metals in pavement joints. Based on sorption

processes with humic substances, and an overall alkaline pH milieu, especially Cu, Cd and Pb, showed low potential mobility and strong adsorption tendency, which could lead to an accumulation and fixation of heavy metals in pavement joints. As the presence of heavy metals in pavement joints poses a direct risk for urban environments, and may also affect environments out of urban areas, if drainage transports accumulated heavy metals. Finally, we encourage further research to give more attention to this special field of urban anthropogenic materials and potential risks for urban environments. Overall urban geochemical

background values, and the consideration of runoff related transport processes on pavements, are needed to develop effective management strategies of urban pavement soil pollutions.

**Keywords.**

Heavy metals, Pavement joints, Urban soils, Urban stormwater, Pollution

## 1. Introduction

The study of heavy metals as environmental pollutants, and their effect on different ecosystems as well as organisms, forms a major research field in environmental science (Alloway, 2013; Blume et al., 2016; Blume et al., 2011; Craul, 1999). In contrast to other pollutants (e.g., organic pollutants) heavy metals are far more widespread, as they are natural components of the earth's crust (Alloway, 2013). Anthropogenic activity, especially mining, industrial processes, traffic and transport, have led to a global increase of heavy metal concentrations in different environmental media like water, air and soil (Cai et al., 2015; Hakanson, 1980; Kowalska et al., 2018; Strode et al., 2009). Along with several emerging threats to the environment, the occurrence and behaviour of heavy metal contamination in soils poses significant challenges for soil ecosystems. Next to the extreme consequences of heavily contaminated soils (e.g. "dead" soil in former mining areas or industrial sites), the presence of heavy metals poses a risk for environmental security, food production, soil organisms and human health (Gałuszka et al., 2014; Strode et al., 2009). It is in this context, and the long-term research on heavy metals in soils, many of today's management practices have become established. Taking Germany as an example, various regulations and laws deal with the topic of heavy metals, and provide recommendations or legislated limits regarding concentrations for soils (e.g., Federal soil protection ordinance) (Bundesregierung, 1998; Blume et al., 2011).

In addition to the natural occurrence of heavy metals in soils, urban areas and their soils are particularly exposed to anthropogenic heavy metal sources. Emissions from industrial or home incinerators, traffic, garbage and construction materials could all be seen as the major sources of heavy metals in urban areas (Gunawardena et al., 2015; Craul, 1999; Manta et al., 2002; Sansalone et al., 1998; Defo et al., 2017; Lu and Bai, 2010; Mahanta and Bhattacharyya, 2011). Although it is possible to distinguish between point sources (e.g., industrial exhaust fumes) and diffuse sources (e.g., brake abrasion, corrosion), urban areas are often very densely built up and heavily exploited, resulting in extensive contamination throughout the area (Manta et al., 2002). Therefore, it is not surprising that the contamination status has become an important key feature of urban soils (Lehmann and Stahr, 2007). In contrast to the extensive research of heavy metal contamination in soils generally, the number of studies specifically investigating urban soils is still small (Burghardt et al., 2015; Schad, 2018).

Pavement joints, defined as the joint space between two or several pavement pieces and filled often by gravel, sand and organic material, fullfill important functions in urban environments. Basically, the question arises whether this material could be named "urban soil", "urban soil material" or must be seen as an anthropogenic material without soil properties. In terms of the World Reference Base for Soil Resources (WRB) update in 2015 as well as the work on urban soils from Burghardt (1995), the pavement joint material could be defined as a part of the todays Technosol reference soil group or as a part of urban soils in general (Burghardt, 1995; Burghardt et al., 2015; Schad, 2018; IUSS Working Group, 2015). In contrast to this the pavement

joint material and its anthropogenic origin differs from soil in its common definition in the most cases (e.g., soil functions or soil development) (Blume et al., 2016). Regardless this question of definition, pavement joints are a common feature in urban areas, like the paving on sidewalks, parking lots and access roads, and are used as a design element in public places. Compared to full sealing, they offer the advantage that infiltration is ensured, which plays an important role in the management of urban runoff (especially stormwater runoff) (Sorme and Lagerkvist, 2002; Sansalone et al., 1998; Dierkes et al., 2005). In this context, they could also be seen as the only anthropogenic material which assumes important functions in extremely sealed inner-urban areas, like the interaction with the atmosphere or as substrate for lower plants (Munzi et al., 2014; Seaward, 1979; Wang et al., 2020). Beside these important functions of pavement joints, a pollution of joints material could pose several risks for the environment and humans. On the one hand, direct exposure to humans (e.g., playing children) or indirect exposure (e.g., urban dust) are thinkable. On the other hand, the potential transport of heavy metals from pavement joints to urban environments through runoff, could affect water ecosystems or even agricultural land (e.g., deposition in floodplains during floods) far outside the urban centres. For these reasons a pollution assessment of pavement joints becomes very important.

If one considers the pollution of urban soils or pavement joint material, however, then urban water management must also be included, because with surface and subsurface runoff, sealed or partially-sealed surfaces and urban soils become a source of pollutants and may pollute urban waters. Therefore, a number of studies have focused on the water quality of surface runoff from urban areas that have been sealed or partially-sealed (Drake and Bradford, 2013; Drake et al., 2014). Sorme and Lagerkvist (2002) examined wastewater, and Sansalone et al. (1998) researched urban roadway stormwater, with a focus on heavy metals. With a focus on urban water and effluent flow, Gilbert & Clausen (2006) studied drainage from different road surfaces, and Wessolek et al. (2009) examined drainage and pollution in sealed areas (Gilbert and Clausen, 2006; Wessolek et al., 2009). In summary their findings show, that surface runoff generally has high concentrations of heavy metals, hydrocarbons and further trace elements. However, the studies so far focused on parking areas or roads, and few studies mention the pavement joints of sealed or partially-sealed surfaces as an urban interface, which can act as a source or sink of heavy metals (Dierkes et al., 1999; 2004; 2005). Furthermore, the pollution retention capability of pavement joints has mostly been determined in laboratory tests and not in the field (Fach and Geiger, 2005). Thus, the understanding of their long-term capacity to retain heavy metals is still limited (Zhang et al., 2018). Apart from this, the question of whether already installed and used pavement joints, not only in car parks but also, for example on pavements, contain accumulated heavy metals, still remains unclear (Dierkes et al., 2005).

In this paper, we report on a case study in the inner-city area of Marburg (Hesse, Germany). The goals of our study were (1) the implementation of a heavy metal pollution assessment of pavement joints distributed in an inner-city area, which to our best knowledge represents the first assessment considering diverse installed pavements and different sampling sites, and (2) to empirically depict possible sources and mobilities of heavy metals in pavement joints with a geospatial approach. The results of this study should improve the understanding of the heavy metal contaminations in pavement joints, which is necessary for the development of targeted urban land management strategies.

## 2. Material and Methods

### 2.1 Study area

Our case study was conducted in the inner-city area of Marburg (Hesse, Germany) located 75 km North of Frankfurt. The city of Marburg covers an area of 123.91 km² including suburban areas (Hessisches Statistisches Landesamt, 2019) (Figure 1). With 67,851 inhabitants (620 persons/km²; Hessian average: 297 persons/km²) the city of Marburg is the eighth-largest city in Hesse. The city is a central town in a rural region. Land use is divided into settlement area (14.9 %), traffic areas (7.5 %) and vegetation containing green spaces, forests and agricultural area (76.5 %) (Hessisches Statistisches Landesamt, 2019). In contrast to the greater city area, the inner city consists of a medieval town centre with dense urban structures, surrounded by residential, university and commercial districts. Traffic in the inner city is composed of local public transport, delivery traffic and individual traffic. Various main streets with high traffic volumes, especially during rush hours, alternate with traffic-calmed zones and squares. Unfortunately, the data base on traffic counts is very limited. A traffic census conducted by the "Hessen Mobil - Road and traffic management" agency in 2015, counted 44,195 vehicles/24h (national highway B3, which passes the city from north to south), 11 728 vehicles/24h (main road L3125, located in the south of Marburg) and 6 145 vehicles/24h (main road L3092, located in the northwest of Marburg) (Figure 1) (Hessen Mobil, 2015). All census points are located at city area entry roads. A second census conducted by the city administration in 2019 counted 13 039 vehicles/24h for an inner-city main road (Ketzerbach, along with sampling site KB) (Bürgerversammlung Marbach, 2019). Out of this limited dataset the traffic volume in the city area of Marburg could be regarded as moderate compared to similar sized cities. However, by concentrating traffic on certain main routes (resulting from urban development and the locations of the main employers) the traffic volumes reach levels above the limits at rush hours.

For the investigation of the spatial variability in the presence of heavy metals in pavement joints, we based our selection of possible study sites firstly on the traffic volume, and secondly on the location of other possible sources of heavy metals. As traffic emissions, and especially brake abrasion or exhaust fumes, are regarded as the main sources of heavy metals on road sites, traffic volume and shielding from roads are important factors for limitation of heavy metal emission (Duong and Lee, 2011; Gunawardena et al., 2015). However, other sources like house emissions, runoff and weathering of urban installed materials (e.g., pavement itself), should also be considered as possible sources (Gunawardena et al., 2015). Based on this, we aimed to investigate sites that exhibited: 1) differences in traffic volume, 2) difference in the distance to roads and 3) are representative of the different types of paving and construction used within the inner-city zone.

Following these criteria, six sampling sites were chosen across the inner-city zone (Figure 1; Figure S2): "Ketzerbach" road (KB) and "Zwischenhausen" road (ZH) as parallel streets, while KB is an arterial road, and ZH is a side road with very low traffic volume, in a traffic calmed area. "Elisabethkirche" (EK), a square around a medieval church with no direct traffic, but exposed on two sides to streets which carry a heavy volume of traffic. The "Audimax" (AM) square and the parking area at the "Schwanhof" (FV), are private places without traffic, but are also exposed to streets. Finally, we chose the site "Marktplatz"

(MP), which is the centre of the medieval district of Marburg, in a traffic calmed area with a high volume of pedestrian traffic.

130    The site MP is located on a hillside clearly above the main traffic routes and completely shielded by buildings.

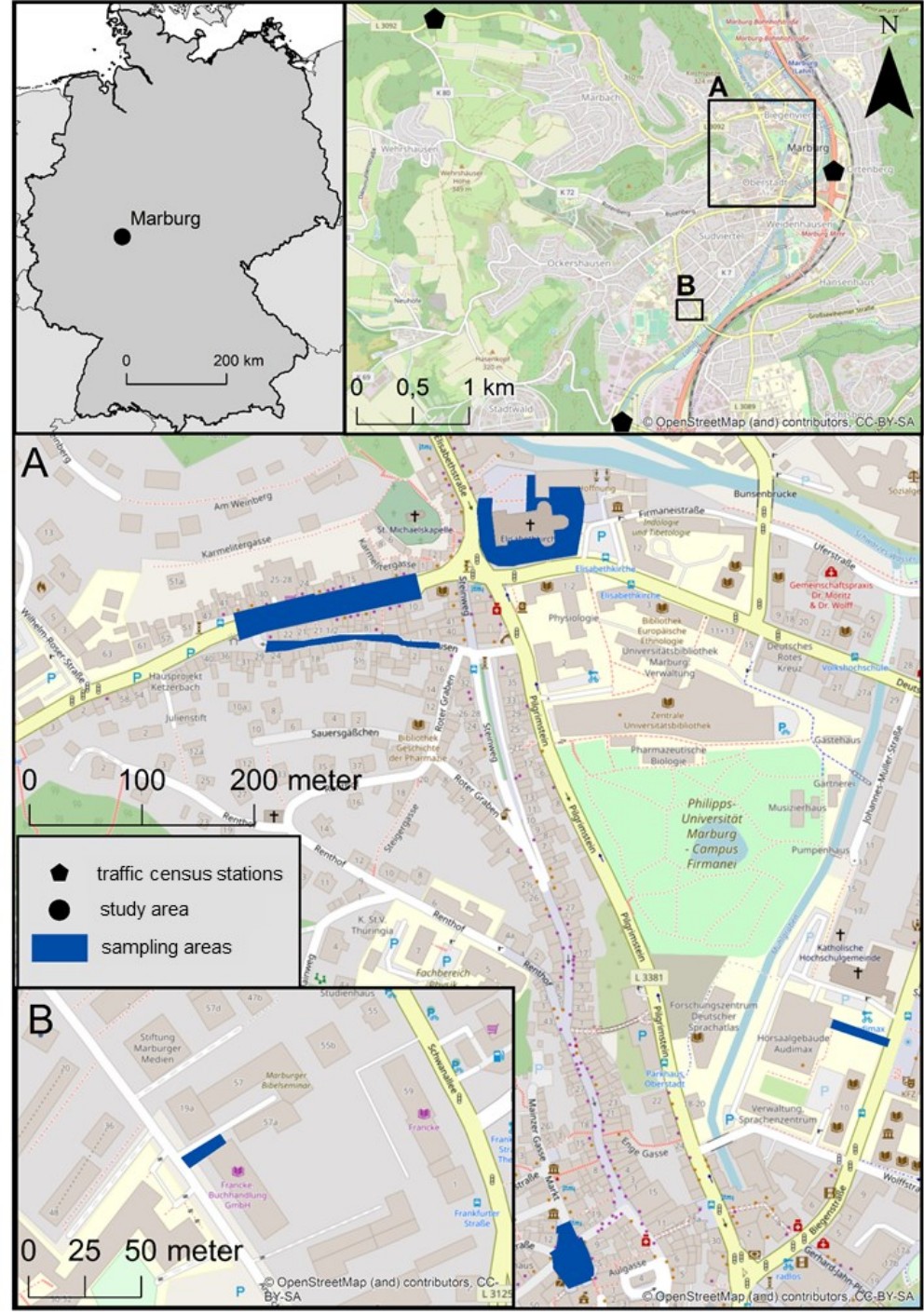

**Figure 1: Study area map composition. A: Inner city area of Marburg. B: Sampling site FV in the south-west area of inner city. Data source: © OpenStreetMap contributors 2020 and Hessian Administration for Soil Management and Geoinformation. Distributed under a Creative Commons BY-SA License.**

## 2.2 Pavement joint material sampling

This study aims to investigate the spatial variability of heavy metals in urban pavement joints, on six study sites located in the inner city of Marburg, and selected according to the above-mentioned criteria (Table 1). At each site, five composite pavement joint material samples from a 1 m² pavement area were taken out of the pavement joints (Figure S1). The sampling was carried out with a metal spatula (stainless steel) and plastic spoon. Material was collected from the joints to a depth of 2 to 7cm in 5 places on each 1m² site. Each composite sample was stored in airtight plastic (PE) bags until further analyses. Sampling points were selected randomly, with the aim of covering the respective sampling area. At site FV and AM the sampling points follow a straight line between street and the next main building. Distance to the nearest road was measured during field work. Material structure in pavement joints was documented according to German soil classification standards (Ad-hoc AG Boden, 2005), and international soil classification standards (FAO, 2006; IUSS Working Group, 2015). In addition, the joint size and the size and type of installed stones were determined. Slope, potential runoff accumulation and absolute heights (metres above mean sea level) were determined by field measurement and additional height data from LiDAR measurements (Hessian administration for land management and geoinformation, 2019). Potential runoff accumulation was classified according to: 0 = no accumulation (slopes > 2°, highest point at site), 2.5 = moderate accumulation (slope < 2°, no specific feature) and 5 = high accumulation (no slope, lowest point at site, drainage gutter or discharge way). Furthermore, the vegetation coverage of pavement joints by mosses, lichens and small vascular plants, was classified (classes: no vegetation = 0; very low coverage < 1% = 1; low coverage < 2% = 2; medium coverage < 10% = 3; strong coverage < 50% = 4; very strong coverage > 50% = 5). The distance from each sampling point to the next traffic frequented street (distance to next road) was measured during field work. Substructure was finally determined by the removal of individual paving stones.

## 2.3 Laboratory analyses

All soil samples were oven-dried at 105 °C for 24 hours. Afterwards each sample was ground by mortar and sieved through a 2 mm stainless steel mesh. The content of organic matter (OM) was measured by loss of ignition (DIN ISO 19684-3:2000-08) (Deutsches Institut für Normung e.V., 2000). The pH value was determined in potassium chloride (KCl) with a pH 91 electrode (WTW, Weilheim, Germany) in accordance with DIN ISO 10390:1997-05 (Deutsches Institut für Normung e.V., 2000). Pseudo-total concentrations of the metals (Al, Fe), heavy metals (V, Cr, Co, Ni, Cu, Sn, Cd, Pb) and the metalloid As, was performed after digestion of 1 g prepared subsample with 20 ml aqua regia (12.1 M HCl and 14.4 M $HNO_3$, ratio 1:3) (DIN ISO 11466:2006-12). Metal content was quantified with an ICP-MS (X Series 2; Thermo Fisher Scientific; Bremen, Germany). The ICP-MS system was calibrated with a certified multi-element standard solution (ROTI®Star, Carl Roth GmbH, Karlsruhe, Germany). Each digest of a soil sample was measured three times, and averaged. The resulting mean metal concentrations were converted into the unit mg/kg. Relative standard deviation (RSD) was quantified for all single measurements, after

threefold measurement to account for data reproduction and effects of heterogeneous matrixes (Weihrauch, 2018; Voica et al., 2012). Data with an RSD $\geq$20% were excluded from further evaluation (Thomas, 2001).

**Table 1 Sampling site and pavement joint features.**

| Sampling Station | Site Features | | | Sampling points | Distance to street (m) | Pavement features | | | | |
|---|---|---|---|---|---|---|---|---|---|---|
| | Usage | Traffic | Slope | | | Pavement material | Average joint size (cm) | plant coverage | pH (KCl) | OM[a] |
| EK | Pedestrian, square around church | not direct frequented, but surrounded from strong frequented streets | medium (2° - 5°) | 1 | 57.02 | sandstone paving stones (50x37 cm) on basalt gravel | 0.87 | 0 | 6.54 | 10.30 |
| | | | | 2 | 55.42 | | 1.40 | 4 | 6.31 | 9.84 |
| | | | | 3 | 21.48 | | 1.26 | 2 | 5.64 | 14.66 |
| | | | | 4 | 15.49 | | 0.91 | 3 | 7.18 | 9.85 |
| | | | | 5 | 13.82 | | 0.86 | 2 | 6.95 | - |
| AM | Pedestrian, square around university building | not direct frequented, single strong frequented road besides | medium (2° - 5°) | 1 | 4.17 | concrete paving stones (40x40 cm) on basalt gravel and sand | 0.90 | 0 | - | 4.74 |
| | | | | 2 | 6.14 | | 0.45 | 1 | - | - |
| | | | | 3 | 8.60 | | 0.55 | 1 | 6.91 | 11.12 |
| | | | | 4 | 30.56 | | 0.55 | 1 | - | 19.44 |
| | | | | 5 | 56.00 | | 0.30 | 0 | 7.52 | - |
| ZH | Pedestrian and traffic, side road | direct frequented, traffic calmed area | low (0° - 2°) | 1 | points direct on street | sandstone paving stones (25x20 or 35x20 cm) on basalt gravel and sand | 1.06 | 1 | 7.35 | 3.55 |
| | | | | 2 | | | 0.83 | 1 | 7.81 | - |
| | | | | 3 | | | 0.93 | 0 | 7.60 | - |
| | | | | 4 | | | 0.94 | 0 | 6.99 | 2.00 |
| | | | | 5 | | | 0.93 | 1 | - | 3.29 |
| KB | Sidewalks at main road | Strong frequented road besides, 13,039 vehicles (24h. 2019) | low (0° - 2°) | 1 | 6.84 | sandstone paving stones (25x20 or 35x20 cm) on basalt gravel and sand | 1.20 | 3 | 6.44 | 5.94 |
| | | | | 2 | 6.89 | | 1.27 | 4 | 6.94 | 10.55 |
| | | | | 3 | 4.66 | | 1.47 | 3 | 6.80 | 3.02 |
| | | | | 4 | 11.00 | | 1.41 | 4 | 7.16 | 5.41 |
| | | | | 5 | 8.90 | | 1.43 | 3 | - | 5.66 |
| MP | Pedestrian, historic marked square | direct frequented, traffic calmed area | low (0° - 2°) | 1 | 21.46 | sandstone and basalt paving stones (heterogenous) | 0.57 | 1 | 7.53 | 7.03 |
| | | | | 2 | 19.83 | | 1.28 | 1 | 7.23 | 8.41 |
| | | | | 3 | 9.50 | | 0.86 | 1 | 7.97 | 5.19 |
| | | | | 4 | 9.70 | | 1.23 | 2 | 7.32 | 6.01 |
| | | | | 5 | 7.35 | | 1.47 | 1 | 6.97 | 9.49 |
| FV | Delivery traffic, carriage entrance company | Individual traffic only, side road ahead | medium (2° - 5°) | 1 | 3.41 | concrete on basalt gravel | 0.34 | 1 | 6.20 | 32.69 |
| | | | | 2 | 8.36 | | 0.41 | 1 | 6.48 | 28.14 |
| | | | | 3 | 14.04 | | 0.35 | 0 | 9.44 | 12.09 |
| | | | | 4 | 18.55 | | 0.31 | 1 | 4.01 | 38.08 |

[a] OM = organic matter (mass %)


## 2.4 Statistics and data evaluation

Basic statistical operations were carried out using Microsoft Excel 2016 (Microsoft, Redmond, USA), R (R Core Team, 2019) and RStudio (Version 3.5.3; RStudio Inc.; Boston, MA, USA). Additional analyses of height data from LiDAR measurements were carried out with ArcGIS (Esri, Redlands, USA) and QGIS (QGIS Development Team).

A major problem during the risk assessment for urban soils and pavement joint material, is the absence of urban geochemical background values of heavy metal contamination. Natural soils and anthropogenic urban soils are only hardly comparable in terms of soil development or soil processes (Burghardt et al., 2015). Out of this circumstance, we decided to perform a three-step risk assessment based on (1) comparison of heavy metal concentrations given in mg/kg with worldwide values, (2) comparison with the available legal requirements and (3) the comparison with regional geochemical background values from

natural soils and soil substrate. As a first approach, however, due to the lack of comparative values, there is only the possibility to attempt an approximation, also in order to enable comparisons with other studies. For step one we took the worldwide general levels of heavy metal and metalloid concentrations for soil surface horizons (SHW, Kabata-Pendias, 2011) and the composition in the upper continental crust (UCC, Rudnick and Gao, 2003). The legal precautionary level for residential areas (LPL) and sandy substrate of natural soils (without land use differentiation) given by the German Federal Soil Protection

Ordinance (Bundesregierung, 1998; Bund-/Länderarbeitsgemeinschaft Bodenschutz, 2003) were applied in step 2. Even if these values are comparatively old, they are nevertheless valid according to the legislation in Germany. Finally, we compare our data with regional geochemical background values from natural soils. Even though the pavement material is not comparable to natural soil, natural materials are used in construction. Out of this reason, we have examined the materials used for most of the pavement joints in the study area: These materials are firstly the substructure of basalt gravel (origin:

Vogelsberg, Westerwald mountains; alkaline basalt) and sand for the first filling of the pavement joints (origin: sand and gravel pits, Lahn valley). From this, we have calculated a geogenic background value (GBH), by averaging the background values for soils from regional volcanogenic substrates (n = 94) and external sand substrates (n = 64) for topsoil and subsoil in Hesse (Friedrich and Lügger, 2011).

Besides the description of the spatial distribution based on basic data evaluation, we have calculated the exposure factor (ExF)

according to Bąbelewska (2010) Eq. (1):

$$y = \sum \frac{Cn - C_{av}}{C_{av}} \tag{1}$$

where Cn—content of heavy metal at a sampling point and Cav—average content of heavy metal at the sampling site. This index is based on the absolute measurement data without the inclusion of geochemical background contents like other heavy metal pollution indices and provides information where, in a given study area, the highest metal loads are located (Kowalska

et al., 2018). The ExF values were compared with absolute terrain heights given in meter (a.s.l.) to proof possible metal transport and accumulation through surface runoff on pavements.

To further investigate the possible accumulation of heavy metal and their spatial variability in pavement joints, we tested for different correlations between our dataset. Spearman correlation analyses, and tests for normal distribution (Shapiro-Wilk test) were performed with the R-packages "graphics", "stats" and "corrplot" (R Core Team, 2018; Wei and Simko, 2017). Interpretation of significant ($p \leq 0.05$) Spearman correlation coefficients (rSP) was carried out according to the following criteria: weak (rSP 0.4 – <0.6), clear (rSP 0.6 – <0.8), and strong (rSP >0.8) (Zimmermann-Janschitz, 2014).

## 3. Results and Discussion

### 3.1 Pavement joint properties

Pavement joints are common in urban as well as suburban areas; they are purely anthropogenic in origin, and therefore linked to human settlement. The build-up material of pavement joints consists of mineral components (gravel, sand), organic components (organic matter) and artefacts (glass, waste, plastics). Based on their properties it could be distinguish between the substructure of pavements and the anthropogenic material in pavement joints itself, which can be considered as an upper pavement structure.

The pavement joints in the inner city of Marburg are mainly built up sands with basaltic gravel. From the surface, starting with a thin layer (< 0.5 cm) of organics (often mosses, lichens, single plants) and organic material, the major part is built up of sands or sandy loams, with organic material and artefacts (waste fragments, glass fragments). A layer of sand or basaltic gravel (partly mixed) follows further down, with concrete or mortar in a few places. Even if the pavement material or substrate could not be classified as a soil in common understanding, the WRB (2015) provides the possibility for classification of those materials. According to WRB classification system, each pavement joint could be classified as: Ekranic Urbic Isolatic Technosol (Arenic, Humic) or Linic Urbic Isolatic Hyperskeletic Technosol (Arenic, Humic) (IUSS Working Group, 2015). Overall, the material built up in pavement joints is very young and have not been undisturbed by conversion measures for long periods.

The average lateral joint size is 0.89 cm (± 0.15) by a vertical depth of 2 – 10 cm. The most common pavement material installed is sandstone (natural pavestones), followed by concrete and basalt paving in different size ranges (Table 1). Plant coverage in joints is heterogenous and occurs in small or medium coverage classes (Table 1). Thus, organic matter varies widely, with a total average of 11.10% by mass; with a standard deviation of 9.25%. Maximum values above 30.0% by mass, occur at single points at sampling station FV. The extent of variability is explained by the variety of growth upon pavement joints, the building age of pavement areas and the joint size. For example, wide pavement joints with a massive growth of moss and other vegetation have a high OM content. However younger, smaller joints, filled only with sand or brash, show overall less OM content. As pavement joints are affected by dust and partly even light plant growth, higher organic matter content is typical like in urban soils (Lehmann and Stahr, 2007).

The pH in pavement joints ranges between 4.01 and 9.44 (average: 6.97). These overall neutral, to slight or medium alkaline properties, can be traced back to the general surrounding of alkaline materials (e.g., plaster itself, concrete) (Räsänen and

Penttala, 2004; Björk and Eriksson, 2002). Additionally, the basaltic underbed could be another factor, as regional basaltic rocks are highly alkaline (Jung and Masberg, 1998). Like OM enrichment, this pH range is specific to urban soils (Lehmann and Stahr, 2007).

In general, these young anthropogenic paterials, which fulfil partly the characteristics of urban Technosols appear in a wide variety, resulting from different materials and construction conditions (e.g., size, substructure). Overall, pavement joints individually are small in size, but through their widespread occurrence they account for a large proportion in urban areas, especially as the only interface between the atmosphere and urban substructure (e.g., in heavily sealed areas).

## 3.2 Heavy metal pollution of pavement joints

All eleven metals studied were detected in each of the samples taken, and analysed via ICP-MS. In general, relative standard deviation of metal concentrations ranging between 0.71 % and 2.84 %, indicates that our data is clustered around mean values with a small overall variation. Only the content of the trace metals Al (35853.00 mg/kg) and Fe (65968.50 mg/kg), reached absolute maximum values. Both elements are ubiquitous in each of the soil and rock materials. In addition, the high values of Al and Fe are a result of the fact, that the under bedding of each pavement is basaltic grit, which when weathered, releases Al and Fe at an increasing rate through decay of the rock (Bain et al., 1980).

From the eleven metals measured, six heavy metals (Cr, Ni, Cu, Sn, Cd, Pb) and the metalloid As will be included in the results analysis as they show comparatively high values. Additionally, global and local geochemical background values for natural soils and legal comparison values are available for these metals. Primarily the concentrations decrease from Sn over Ni>Cr>Cu>Pb>As to Cd (Table 2).

Starting with the general comparison on a global level average and median concentrations of each heavy metal exceed the average concentration in natural soil surface horizons worldwide (Kabata-Pendias, 2011) (Table 2). This circumstance is also based on the fact that the heavy metal concentrations exceed also the assumed metal concentration in the upper continental crust (Rudnick & Gao, 2003). In contrast, the As concentration is significant higher than average SHW value but lower than the UCC value. These comparisons are only possible to a limited extent, as these are very general global mean values, but necessary as proper background values for pavement joints or urban anthropogenic substrates and soils are lacking. A better approximation can be achieved by comparing the heavy metal concentrations with the geochemical background in hessian soils (Friedrich & Lügger, 2011). The average concentration of Cr, Ni, Cu, Cd and Pb exceeds the calculated GBH values, whereas As stays below the value. Summarized the heavy metal loads in pavement joints are on average 3.6 (SHW), 4.4 (UCC) and 2.9 (GBH) times greater than the comparative values and in the case of Sn 245.4 times greater than SHW and UCC.

This exceeding is generally not surprising, as each sampling site is exposed to different anthropogenic heavy metal sources. Besides the release of heavy metals, either by dust or gas in urban areas through incineration, there are other traffic associated sources (Bryan Ellis and Revitt, 1982; Duong and Lee, 2011). For example, Cd, Cr and Ni as a product of combustion of fossil fuels, can reach pavement joints through emissions (Duong and Lee, 2011). Pb, Cu and Sn are strongly associated with traffic emissions (fossil fuel), or the abrasion from brake pads and tyres (Duong and Lee, 2011; Yan et al., 2013). For the spatially

narrow inner-city area, both larger point sources and a large number of diffuse sources, must be considered with this in mind, the highest concentrations of heavy metals coincided with typical urban source patterns. Although increased contamination is typical for urban soils (Schad, 2018; Lehmann and Stahr, 2007), partial concentrations can also be attributed to the materials used in pavement construction. Jung & Masberg (1998) noticed high concentrations of Ni, Cr and Co for mafic volcanic rocks from the Vogelsberg mountains located next to the city of Marburg, and with important quarries for regional construction activities (Fach and Geiger, 2005).

In contrast to this significant exceedance of geochemical backgrounds, the heavy metal concentrations do not exceed any of the legal precautionary levels for residential areas (Bundesregierung, 1998). Only the maximum values of Cr and Ni are above the given LPL values, as well as the third quartile value of Ni (168.3 mg/kg) and therefor approximately 25 % of the samples. As the given LPL values are the legal basis for the need of action in case of soil contamination in Germany, the data indicate that there is no need for countermeasures on a legal basis. However, as the LPL values are on average 9.6 times greater as the mean values of Cr, As, Cd and Pb in pavement joints, it seems to be difficult to achieve a status where action must be taken at the legal level. Apart from this taking into account other values of the legislation without a direct need of action, like the preventative-values for natural soils or preventative thresholds for sandy substrates, both values are exceeded by the average concentrations of Cr, Ni, Cd and Pb (Bundesregierung, 1998). Furthermore, the range of Cr and Ni concentrations were above the preventative values for playgrounds (200 mg/kg Cr; 70 mg/kg Ni; Bundesregierung, 1998) at 5 sampling points for Cr (27 sites for Ni respectively). Irrespective of the fact that there is no need for countermeasures at the legal level, the fact that the comparative values and less strict legal values were exceeded indicates that a potential risk for humans as well as urban and extra-urban environments cannot be excluded. Further discussion about a first risk assessment for pavement joints is given in chapter 3.5.

**Table 2** Summary of metal and heavy metal concentrations (given in mg/kg) in pavement joints compared to natural background levels and legislation values.

| n = 29 | | Cr | Ni | Cu | Sn | As | Cd | Pb |
|---|---|---|---|---|---|---|---|---|
| | | | | | mg/kg | | | |
| Mean | | 157.1 | 160.2 | 101.7 | 560.2 | 3.9 | 1.2 | 61.4 |
| Median | | 109.7 | 140.3 | 94.4 | 387.6 | 4.0 | 1.1 | 50.9 |
| Min. | | 43.4 | 48.5 | 35.7 | 104.6 | 1.3 | 0.3 | 2.9 |
| Max. | | 1290.3 | 474.3 | 252.5 | 3728.1 | 7.6 | 4.1 | 187.7 |
| Standard deviation | | 220.6 | 97.4 | 54.9 | 647.1 | 1.4 | 0.8 | 43.5 |
| SHW [a] | Average content surface horizons worldwide | 60.0 | 29.0 | 38.9 | 2.5 | 0.67 | 0.41 | 27.0 |
| UCC [b] | Composition in upper continental crust | 92.0 | 47.0 | 28.0 | 2.1 | 4.8 | 0.09 | 17.0 |
| GBH [c] | Geochemical background in Hessian soils | 68.8 | 101.5 | 26.3 | - | 5.7 | 0.19 | 25.5 |
| LPL [d] | Legal precautionary level for residential areas | 400.0 | 140.0 | - | - | 50.0 | 20.0 | 400.0 |

[a] Kabata-Pendias (2011); [b] Rudnick & Gao (2003); [c] Calculation explained in method section according Friedrich & Lügger (2011); [d] German Federal Soil Protection Ordinance - BBodSchV (1998)


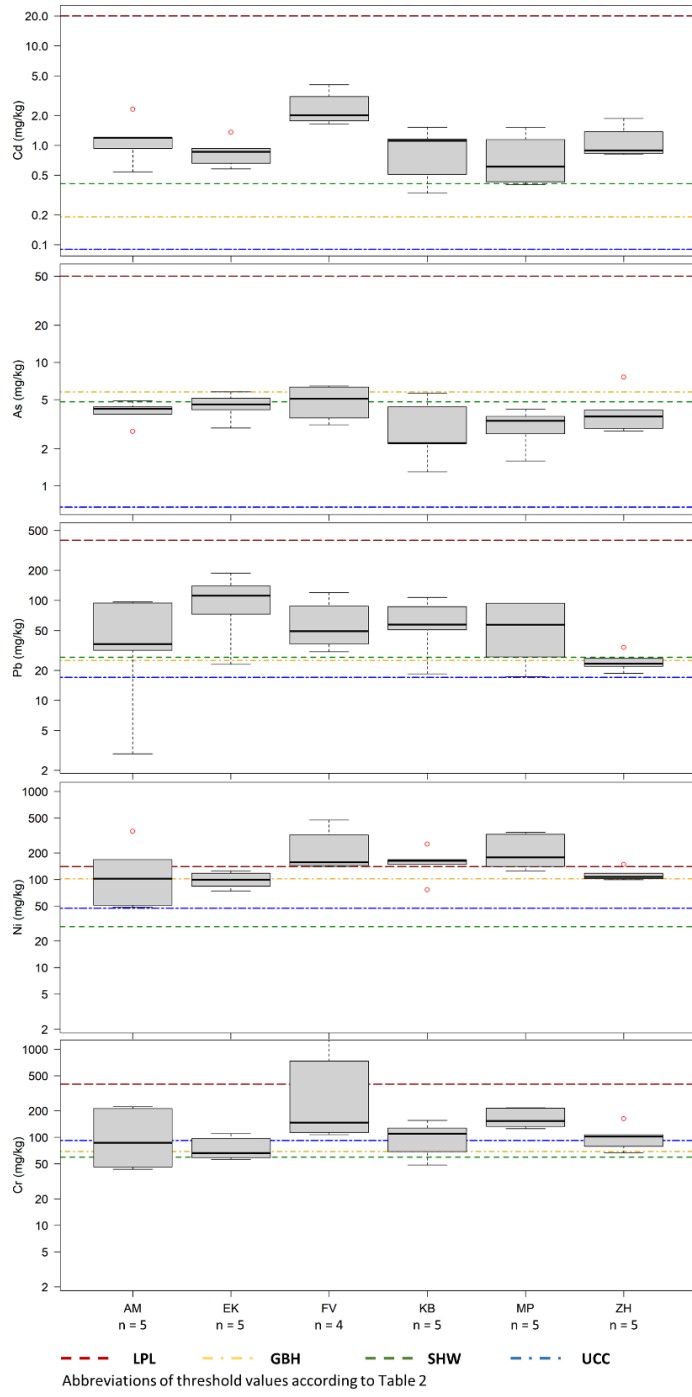

**Figure 2: Concentrations of the heavy metals (given in mg/kg) Cd, Pb, Ni and Cr with the metalloid As according to sampling sites and different thresholds according Table 2.**


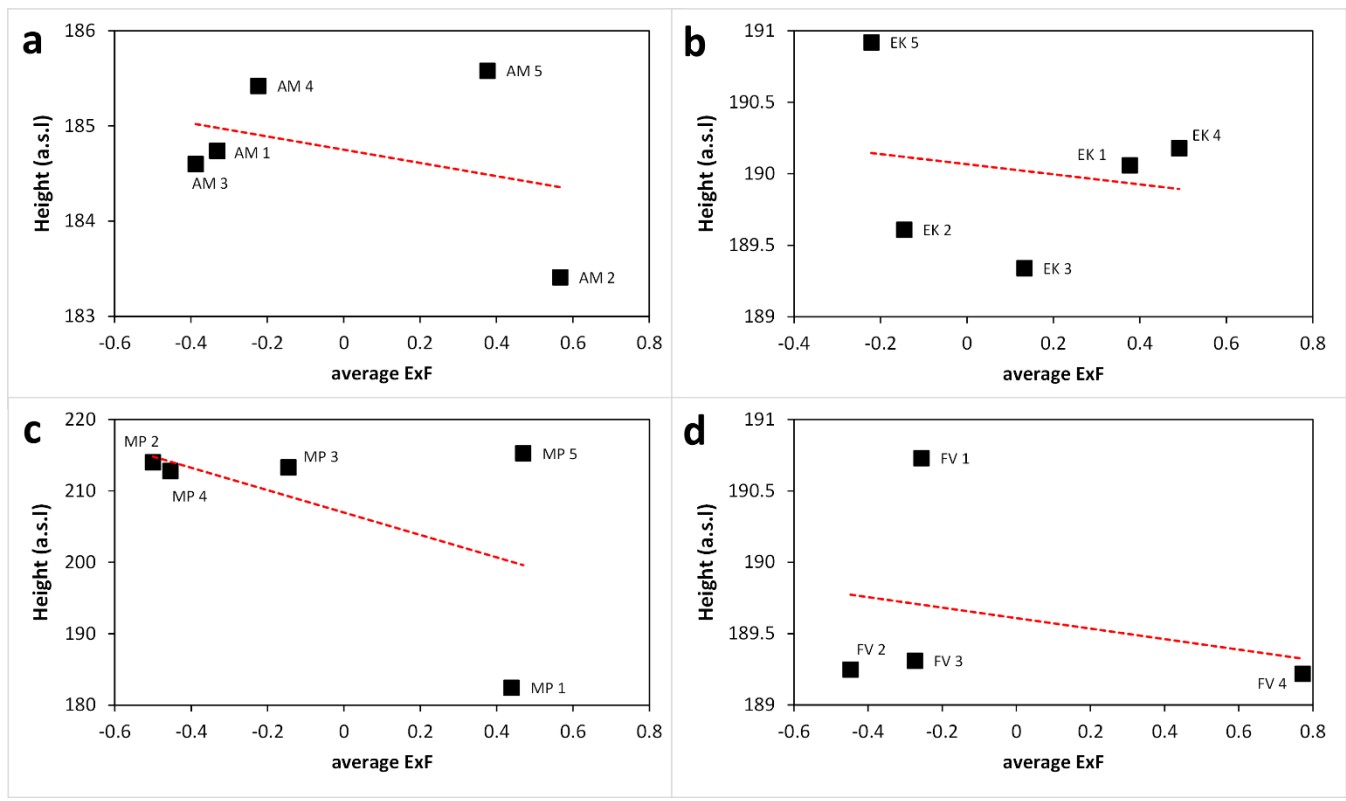

**Figure 3: Exposure factor (ExF) according to sampling point height (metres above mean sea level) for all sampling sites with significant height differences. a) Sampling site AM; b) Sampling site EK; c) Sampling site FV; d) Sampling site MP. Linear trend showed in red dotted line.**

### 3.3 Spatial variability of heavy metal pollution

Different studies dealing with the topic of heavy metal contamination of pavement or urban soils, have noted significant influence of: a) spatial traffic differences (e.g., traffic volume, braking points at crossings) and or b) land use (e.g., industrial sites versus parks) and or c) if considering urban runoff (especially stormwater runoff), the distance to inlets or other emission sources as the major drivers for spatial patterns of urban heavy metal pollution (Bryan Ellis and Revitt, 1982; Duong and Lee, 2011; Herngren et al., 2006; Tedoldi et al., 2017; Logiewa et al., 2020). Through the selection and distribution of our sampling sites and points, we can give results for spatial distribution of heavy metals in pavement joints on two spatial levels: Level one with a differentiation between sampling sites located in the inner-city area of Marburg and, level two including the comparison of single sampling points at each site with its neighborhood and distance to different heavy metal sources.

Comparing the individual metal concentrations on the spatial level one the concentrations of Cd, Pb and Ni exceed the SHW and UCC levels at each sampling site (Figure 2). The average As values are ranging at each site between the SHW and UCC level, whereas the average Cr values exceed both values at four sampling sites. Also, the regional geochemical background for

natural soils is exceeded for all metals (except As) at the main number of sampling sites. Out of these findings, a general pollution of pavement joints in the inner-city area could be concluded.

However, within this general pollution, differences between sampling sites and points occur. Maximal levels of Cd, Ni, Cr as
well as Sn are reached at site FV, followed by MP and AM. Although the median values are quite narrowly spaced, the differences become clear in the maximum values. Considering traffic emissions (a) or a specific land use (b) as main sources of urban heavy metal pollution and accumulation in pavement, it is interesting that the maximum concentrations are reached at FV (sidewalk and private driveway at secondary road) and MP (historic market place, traffic-calmed area). In both cases no particular exposure to traffic or contamination-promoting land use (e.g. industry) can be identified. However, exposure to
traffic could play an important role for sites AM and EK, as both sites, though not used directly by vehicles, are exposed to major roads. In comparing the locations KB and ZH, the heavy metal concentration at KB is 2.5 times the values from ZH. This finding could be traced back to the higher traffic frequency at site KB (major road), in contrast to side road ZH (Manta et al., 2002). Individual differences between the five sampling points at site KB are explainable by stop and go traffic, as levels of Cu, Cd, and Pb are often higher in areas around traffic lights, and Ni as well as Cu are attributed to braking (Duong and
Lee, 2011). The locations EK and MP are traffic-calmed areas, and the highest concentration at EK is reached at EK-3, which is the nearest point to both heavily used streets (Figure S2). The lowest concentration is reached at EK-5, a sample point which is shielded from the streets by the church building, and where traffic emission sources cannot fully reach as Hagler et al. (2011) noted in their study. At site MP the highest level is clearly reached at MP-4 and MP-5. They are the only sample points with higher concentrations. MP-4 is located directly in a rainwater drainage channel and MP-5 is subject to a lot of surface runoff.
At site AM, maximal values are reached for Cr, Ni, Cu and Pb next to the major street (AM-1, AM-2) (Figure S2). As sampling site AM was a linear one, AM-5 (greatest distance to street) also shows higher values, especially for Cd, and the second highest PLI is recorded at this site, whereas the other linear sampling site FV, reflects an alternative trend. At site FV the highest concentration of each heavy metal and pollution index is FV-3, followed by FV-4. Both sites are located furthermost from the road.

Considering maximum heavy metal loads the highest concentration are reached at point FV-3 for Ni (474.3 mg/kg), Cr (1290.3 mg/kg), Sn (>3000 mg/kg) and Cd (4.1 mg/kg) followed by MP-4 and MP-5 and AM-2 with additional higher loads. Critical legal maximum values for Ni (residential area) are reached by 100.0 % of sampling points at FV and 80.0 % at KB and MP (Figure 2). The legal maximum value for Cr (playground) is reached at a maximum of 40.0% of all points at site AM and MP. It could therefore be concluded that single pollution hotspots occur in the inner-city area of Marburg. The origin of these
hotspots cannot be attributed exclusively to traffic, as stated in the majority of other studies dealing with heavy metals in urban soils (Yan et al., 2013; Herngren et al., 2006; Duong and Lee, 2011; Bryan Ellis and Revitt, 1982).

Spearman correlation between the distance from each sampling point to the next traffic frequented street (distance to next road), shows weak positive correlations for Sn and Pb in subordinate data (all sampling sites) (Table 3). Additional single

sampling site specific significant correlations occur at site AM for Cd, at site EK for Ni and site KB for Sn. Beside these results, the correlation between distance to next road and the ExF is not significant for subordinate and specific data. Therefore, it can be concluded that traffic emissions do not seem to be the main reason for the spatial distribution of heavy metals.

Another possible factor in understanding the distribution patterns, could be urban drainage and surface runoff including stormwater runoff, as drainage could be possible transport medium for heavy metals on paved areas (Gilbert and Clausen, 2006; Tedoldi et al., 2017). The sites AM and FV are examples where the influence of slope and drainage for distribution of heavy metals in pavement joints can be monitored. The highest individual metal concentration and ExF values are reached at the lowest section in these areas, and nearby drainage gutters (Figure 3). FV has a consistent slope from FV-1 to FV-3. Only FV-4 is located beyond the gutter on a flat section. Site AM presenting a similar case: sample point AM-2 with higher concentrations, is also the lowest point and nearby the gutter. The samples were taken in a straight line from AM-1 near a road, over a gutter to AM2 and across a permanent incline to a second gutter (AM-5), in front of a building.

Additional Spearman correlation analysis was carried out to test these relationships between potential runoff accumulation and absolute height (above mean sea level) of each sampling point, with metal concentrations and the ExF (Table 4).

Overall, strong positive correlations ($p < 0.05$) were found between runoff accumulation value and the ExF and Cr values. Additional clear positive correlations were found for Ni, Cu, Sn, As and Pb (Table 4). However, the correlation with absolute heights (meters above mean sea level) shows no clear subordinate correlations. For the sites AM, MP and FV a clear trend is apparent when plotting absolute heights against ExF data (Figure 3). Highest pollution loads are reached at the sites clearly at the lowest point. In the case of site MP, the highest ExF value is reached by the sampling point with largest terrain height. As concluded from field work, sampling point MP-5 is higher, but have the highest potential runoff accumulation, as they are reached by a larger surface runoff area above, and located next to discharge points in the pavement or drainage gutters. In the case of site EK-1 and EK-4 with medium terrain height for the site show highest ExF values, which could also be traced back to the location of drainage gutters on site. This result is contrary to other findings, as Tedoldi et al. (2016) reported highest accumulation of heavy metals at inflow points, as a consequence of filtration capacity.

**Table 3 Spearman correlation between distance to next road, Exposure Factor (ExF) and metal concentrations.**

| Variable A | Variable B | All sampling sites | | AMª | | EKª | | KBª | | MPª | |
|---|---|---|---|---|---|---|---|---|---|---|---|
| | | spearman ρ | p | spearman ρ | p | spearman ρ | p | spearman ρ | p | spearman ρ | p |
| | OMᵇ | **0.50** | **0.01** | **1.00** | **0.00** | 0.00 | 1.00 | 0.10 | 0.87 | -0.10 | 0.87 |
| | pH | -0.15 | 0.50 | **1.00** | **0.00** | -0.20 | 0.80 | 0.80 | 0.20 | 0.30 | 0.62 |
| | ExFᶜ | 0.12 | 0.55 | 0.20 | 0.78 | 0.60 | 0.35 | 0.70 | 0.23 | -0.40 | 0.51 |
| | Cr | -0.21 | 0.27 | -0.60 | 0.28 | -0.70 | 0.19 | -0.60 | 0.28 | -0.60 | 0.28 |
| Distance to next road (m) | Ni | -0.27 | 0.16 | -0.60 | 0.28 | **-0.90** | **0.04** | -0.60 | 0.28 | -0.70 | 0.19 |
| | Cu | 0.32 | 0.09 | -0.50 | 0.39 | 0.60 | 0.28 | 0.20 | 0.75 | -0.40 | 0.50 |
| | Sn | **0.52** | **0.00** | 0.70 | 0.19 | 0.60 | 0.28 | **0.90** | **0.04** | -0.50 | 0.39 |
| | As | 0.20 | 0.29 | 0.20 | 0.75 | -0.50 | 0.39 | -0.20 | 0.75 | -0.20 | 0.75 |
| | Cd | 0.07 | 0.73 | **0.90** | **0.04** | 0.60 | 0.28 | 0.80 | 0.10 | -0.70 | 0.19 |
| | Pb | **0.58** | **0.00** | 0.40 | 0.50 | 0.60 | 0.28 | 0.50 | 0.39 | -0.40 | 0.50 |

[a] = Sampling sites; [b] = Organic matter; [c] = Exposure Factor

385

**Table 4 Spearman correlation between runoff accumulation value, Exposure Factor (ExF) and metal concentrations for sampling points at sites significant height differences (sites: AM, FV, MP).**

| Variable A | Variable B | Spearman's rho | p-value |
|---|---|---|---|
| | ExF[a] | 0.80 | 0.00 |
| | Cr | 0.80 | 0.00 |
| | Ni | 0.76 | 0.00 |
| runoff accumulation value | Cu | 0.76 | 0.00 |
| | Sn | 0.65 | 0.01 |
| | As | 0.61 | 0.02 |
| | Cd | 0.35 | 0.21 |
| | Pb | 0.68 | 0.00 |

[a] ExF = Exposure Factor

## 3.4 Accumulation and mobility tendencies of heavy metals in pavement joints

Each sampled pavement joint in the inner city of Marburg shows an alkaline soil milieu and high organic matter (OM) content (Table 1). Additional correlation analyses of OM content and pH with metal concentration, reveals two groups with significant relationships ($p = < 0.05$), but opposite and reversed conditions (Figure 5). Group 1 including the trace metals Al and Fe shows slightly weak negative correlations with OM and weak positive correlations with pH. Accordingly, group 2 including Cu, Sn, As, Cd and Pb shows clear positive correlation with OM and weak negative correlation with pH. The weak negative correlations are consistent with the findings from urban soils were no significant correlation occurs (Mante et al., 2002) Additional clear to strong positive correlations between metals themselves appear between Al, Fe, Cr and Ni. In contrast to very weak negative and positive correlations of this metals with the group of Cu, Sn, As, Cd and Pb. In addition, Cu, Sn, As, Cd and Pb have strong to weak positive correlations among each other. From these findings, based on significant inter-element relationships, the dataset could be divided into group 1 (Al, Fe, Cr, Ni) and group 2 (Cu, Sn, As, Cd, Pb). Other authors suggest, that strong inter-element relationships could be traced back to a combined metal pollution from similar long-term sources (Manta et al., 2002; Lu and Bai, 2010). Therefore group 1 and group 2 metals may have a different origin or controlling factors (Dragovic´ et al., 2008).

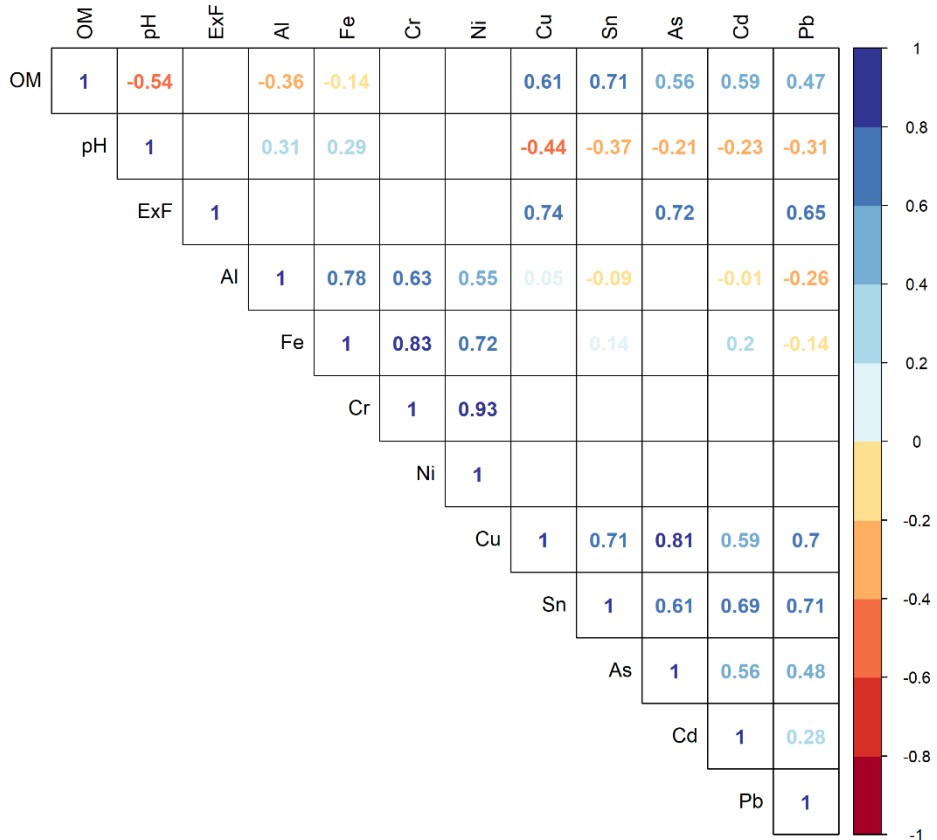

**Figure 4: Spearman correlation matrix for organic matter (OM), pH, Exposure factor (ExF) and elemental concentrations in pavement joints. Spearman's rho displayed in grid if correlation is significant (p-value < 0.05). Positive correlation in blue colours, negative correlations in red colours.**

Next to those inter-element relationships, different correlation clusters with OM and pH could allow suggestions about metal, metalloid and heavy metal retention in pavement joints as both values could be important controlling factors. First of all, the pavement joint substrate, built up from sands and coarse gravel with artefacts, allows only poor adsorption of heavy metals on clay minerals, silt or pedogenic oxide surfaces (Blume et al., 2016; Alloway, 2013). In contrast, the high content of OM could lead to the sorption on humic substances, and the formation of metal-humus-complexes (Herms and Brümmer, 1984; Alloway, 2013). In this context, we interpret the clear to strong positive correlation between group 2 metals (Cu, Sn, As, Cd, Pb) and OM, as evidence of those processes. The adsorption was also found in urban soils for Cu and Pb, but not for Cd (Mahanta and

Bhattacharyya, 2011). Also, Defo et al. (2017) found a major influence of OM on the retention of Pb as well as Cd, in urban soils (Defo et al., 2017). These different findings stayed in line with the fundamental role of OM for Pb and Cu sorption in soils (Lee et al., 1998; McLaren et al., 1983). The overall neutral to alkaline pH milieu (6.97 total average), supports a fixation of metals by specific bindings, since unspecific bonds and dissolutions only occur at more acidic pH values (Herms and Brümmer, 1984; Blume et al., 2016). Comparing the opposite relationships of group 1 and group 2 metals, with additional inter-element relationships, we conclude a medium fixated group of metal-complexes including Al, Fe, Cr, and Ni with less relationships to the controlling factors OM and pH (medium potential mobility, less adsorption tendency), and partwise more fixed group of Cu, Sn, As, Cd and Pb with stronger relationships to the controlling factors (low potential mobility and strong adsorption tendency).

Out from the correlation analyses and the additional widespread spatial pollution with comparatively high concentrations of different metals in pavement joints, an accumulation of metals in pavement joints could be stated. Available sorbents, the alkaline environment and a constant supply of heavy metal emissions from minimal two different main sources, provide suitable conditions for pollution accumulation. This point deserves special attention, as other authors have noted heavy metal enrichment by organic material, especially in gutters, which would demonstrate a link between transport by water and accumulation at runoff gathering points (Bryan Ellis and Revitt, 1982). Even if the major heavy metal share with the metalloid As could be seen as accumulated and more or less fixed by OM and pH conditions, this accumulation can become especially problematic in the case of strong surface stormwater runoff through precipitation. If urban surface or stormwater runoff is regarded as a major driver for heavy metal transport and accumulation at the lowest points, a further transport after the infiltration in pavement joints seems to be possible. Particle uptake and transport as suspended load as major process, as well as the transport as dissolved metals in surface runoff for the less fixed metal group as secondary process could be possible (Gilbert and Clausen, 2006). Applying this on a larger scale, and considering urban pavements as pollution sources for the environment out of urban areas, polluted urban runoff could provide a link between both systems, as stormwater runoff especially is discharged directly into receiving waters from urban areas, less polluted environments like rivers, wetlands and floodplains in downstream areas may also be affected. Therefore, a risk for agricultural land in alluvial zones, as well as for river ecosystems, is conceivable.

### 3.5 Contamination pathways and risk assessment for heavy metal pollution in urban pavement joints

Heavy metals as environmental pollutants could pose a wide variety of potential risks for different ecosystems, plants and animals (Alloway, 2013; Blume et al., 2016; Blume et al., 2011; Craul, 1999). Even if different metals act as important trace metals (e.g. Ni, Co, Cu) for organisms, increased or excessive concentrations in combination with the hazardous and toxic

properties of other metals (e.g., As, Cd) lead to the need of risk assessments for heavy metals in the environment (Alloway, 2013).

Common practices developed for risk assessment of heavy metal pollution in natural soils are based on the consideration of various direct and indirect contamination and exposure pathways (e.g., soil-human, soil-plant) (Blume et al., 2016). Additionally, exposure time and potential doses or uptake quantities are important factors to evaluate potential risks. Furthermore, negative influences on soil functions must be assessed (Gałuszka et al., 2014; Strode et al., 2009).

Often rules and limit values from legislation can be used for this purpose, such as the precautionary values for soils from the

German Federal Soil Protection Act applied in this study (Bundesregierung, 1998; Blume et al., 2011). Various geochemical indices (e.g., Geoaccumulation index, Pollution load index, Ecological risk index) are also applied to assess the potential contaminations (e.g., ecological risk index) (Kowalska et al., 2018):

In case of urban pavement joints and their contamination with heavy metals, common tools and practices need to be reconsidered for a risk assessment. Geochemical indices for natural soils based on geochemical background values, cannot be

applied purposefully due to the lack of data and studies, as there is no basis for comparison. Limit and threshold values based on legislation can be applied, but may underestimate potential risks as inner-urban areas are preferred living areas for humans and lead to long-term exposures to urban pollutants (United Nations, 2019). The combination of expected long-term or live-time exposures and the world-wide increase in urban population, highlights the necessity for a first risk assessment on pavement joints pollution (United Nations, 2019).

Under consideration of the given limitations of this study, we conclude the following main issues as important for a general risk assessment (regarding chapter 3.1-3.4: a) A metal contamination of urban pavement joint material with single spatial hotspots from multiple sources is given, even if legal precautionary levels are only exceeded by single maximum concentrations; b) Spatial distribution of metal concentrations indicates inner-urban transport and accumulation of metals through surface runoff; c) Pavement joint properties and statistical relationships indicate fixation and accumulation tendencies

for heavy metals in pavement joints.

When combining these findings with previous research work, each contamination pathway starts with a potential source for heavy metals in urban areas. Even if this study can only give limited estimations on possible sources (e.g., no direct relationship to traffic emissions), different authors taking various potential sources into account (e.g., emissions, construction material) (Gunawardena et al., 2015; Craul, 1999; Manta et al., 2002; Sansalone et al., 1998; Defo et al., 2017; Lu and Bai, 2010;

Mahanta and Bhattacharyya, 2011). Independent of the specific source, three different processes result from the presented data, which could be responsible for the observed accumulation (Figure 5a). Rainfall followed by surface runoff or infiltration on the one hand, and airflow on the other hand. Starting with processes which require water as transport medium, the transport of metals in dissolved form or as suspension freight is conceivable. The possible flow distance is limited at the surface by the network of drainage units (e.g., gutters, gullies) and the comparatively high infiltration capacity of paved areas. Infiltration

rates from the literature for different urban substrates ranging between 80 cm/h to 2,000 cm/h (permeable interlocking concrete pavers, Bean et al., 2007) or 1.2 cm/h to 577.1 cm/h (pervious concrete, Chopra et al., 2010) and could be named as highly heterogenous based on the structure properties. This transport processes, especially during heavy rainfall events due to the high sealing in inner urban areas, can lead to accumulation and deposition of metals in the pavement joint material, as well as to initial accumulation in puddles or further transport into the urban sewage system. In addition, transport by airflow is

conceivable as a further process, especially under dry conditions. High heavy metal contents in road dust are already known (Christoforidis and Stamatis, 2009; Duong and Lee, 2011; Logiewa et al., 2020). Under dry and windy conditions, pavement joints, which are mostly open and not covered with plants, can theoretically lead to a constant alternation of transport and deposition, which can be intensified by urban wind systems and heat islands (Grimmond and Oke, 1998; Arnfield, 2003; Vardoulakis et al., 2003).

With consideration of these transport or migration processes direct and indirect exposures, additionally divisible by spatial units, could occur for heavy metals in pavement joints (Figure 5b).

For inner-urban areas direct exposure to humans and especially children through oral, inhalative or dermal uptake a conceivable. From the legislation side, no trigger or action thresholds are exceeded by the detected heavy metal concentrations, which indicates just a low exposure risk (Bundesregierung, 1998; Blume et al., 2016). In contrast, the low exposure risk could

be higher than assumed, taking the long-term exposure for humans into account. As pavement joints are primarily built in residential areas, those areas are simultaneous preferred living areas, which makes a daily exposure possible for a lifetime (Luo et al., 2012; Li et al., 2014). The direct exposure against other species like animals or plants is more difficult to assess, since urban areas are not the preferred habitat of most species here. The uptake of metals is basically conceivable, the risk of damage is probably low (Munzi et al., 2014; Wang et al., 2020).

In addition to these potential risks, there are other indirect exposures outside the inner urban area. As a result of the transport processes mentioned above, heavy metals can quickly pass through wastewater into rivers, especially during heavy rainfall events. The consequences of heavy metal inputs into aquatic systems from urban or other sources (e.g., mining sites) are well known (Sansalone et al., 1998; Sorme and Lagerkvist; 2002; Atanackovic et al., 2013). As a result, the quality of the water can deteriorate significantly and there are many negative influences on aquatic organisms (Atanackovic et al., 2013; Sakson et al.,

2018; Zhang et al., 2020). In addition, heavy metals may also reach semiterrestrial systems (e.g., floodplains), their soils and enter further contamination paths (e.g., plant-root uptake) through flooding (Miller, 1997).

The potential contamination pathways and risks posed by heavy metal contamination in urban pavement joints can be described as manifold, which leads to a clear demand for more research in this field in order to conduct precise risk assessments. Based on the present study, we estimate the risk of direct exposures within our study area to be low. Nevertheless, there is also a risk

for non-urban areas which should not be neglected. More intensive use of urban areas during increasing extreme weather events caused by climate change (e.g., heavy rain, flooding, heat) can intensify the risk of heavy metal discharge and the

negative impacts on important aquatic ecosystems (IPCC, 2018). Finally, the question arises whether the risk even for urban ecosystems and urban habitats, as well as for functions of the pavement joints (e.g. infiltration, heavy metal retention) should be reassessed in the future.


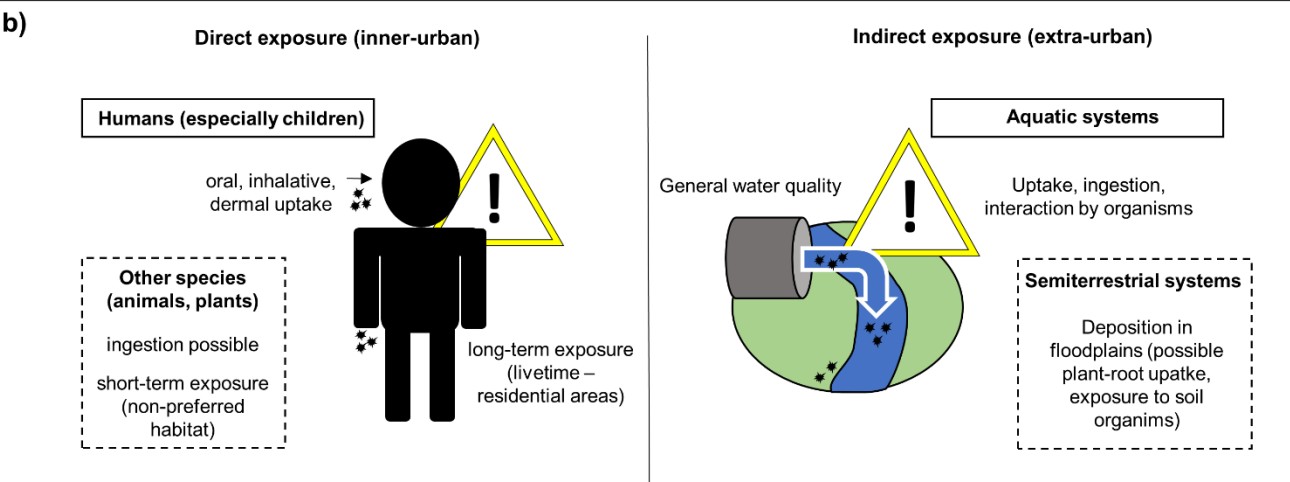

**Figure 5: Risk assessment for heavy metal pollution in urban pavement joints. a) Potential contamination pathways of heavy metals in urban pavement joints; b) Potential risk exposures from heavy metals from urban pavement joints.**

**4. Conclusion**

In our study, pavement joints, mentioned as an important part of the urban soils, were found to be polluted by Cr, Ni, Cu, Cd and Pb at each sampling site, as shown by absolute concentrations. Despite the limitations mentioned, spatial comparison

showed that traffic emissions are not the main cause of the spatial distribution of heavy metals in pavement joints. Instead, we found strong correlations between runoff accumulation and heavy metal pollution, mainly at runoff gathering points. The accumulation of heavy metals at gathering points is supported by an alkaline pH milieu and adsorption processes on organic material, which makes up a substantial part of pavement joints. Therefore, the material used during construction of pavements should be carefully considered, so as to avoid anthropogenic soil environments that foster heavy metal accumulation (basaltic rock material with highly alkaline milieus). As pavement joints are mainly constructed with the function of water infiltration in sealed areas, solution and transport of accumulated heavy metals poses a possible risk for the environment outside of urban areas (e.g., river ecosystems, floodplain systems). In addition to the direct risks of accumulated heavy metals (e.g., direct exposure to humans or urban dust emissions) current research needs to pay more attention to this special field of urban soils. We encourage the following topics to be regarded as relevant for further research:

- More attention should be paid to pollution of pavement joints as well as urban soils in general, as these soils play a major role in urban environments, can react as pollutant accumulative materials and may pose a direct risk to humans.
- More research on urban soil pollution could enable the development of urban geochemical background values for different pollutants, which promote more effective risk assessments and spatial comparisons, even with pollution indices.
- Different sources of heavy metals besides traffic and transport in urban areas (e.g., surface runoff), need to be considered to develop effective management strategies of urban soil pollution.
- The role of the runoff must be examined more closely. Further studies about pollution concentration in drainage from pavement, and infiltration of drainage, are necessary. Not only on single areas, but with spatial (e.g., geospatial sampling approaches) and temporal (e.g., long-time studies, event-based sampling) resolution.

**Data availability**

Our research data is available in the following data repository: Weber, Collin, J.; Santowski, Alexander; Chifflard, Peter (2020), "Spatial variability of heavy metal concentration in urban pavement joints – A case study", Mendeley Data, v1 http://dx.doi.org/10.17632/b3d66r56k8.1

**Author contribution**

Collin J. Weber has carried out the conceptualisation and selection of methodology. Collin J. Weber and Alexander Santowski performed the data curation, investigation and formal analyses. Peter Chifflard carried out the project administration and supervision and provided the resources. Visualisation and writing of the original draft were performed by Collin J. Weber with contributions of all co-authors. Writing during review & editing was performed by Alexander Santowski and Peter Chifflard.

## Competing interest

The authors declare that they have no conflict of interest.

## Acknowledgments

The authors gratefully acknowledge the support during laboratory analyses and ICP-MS measurements through Nina Zitzer.

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
