# Peer review of "Spatial variability of heavy metal concentration in urban pavement joints – A case study"

_SOIL, 2020_

## Referee Comment (RC1) · Anonymous Referee #1 · 6 Sep 2020

Novel and interesting research with the focus on the specific urban soils, typically considered "sealed soils", i.e. impermeable for surface water and hampering the plant rooting, fauna activity, cycling of elements. There is a wide variability of such Technosols, from absolutely impermeable (covered with continuous concreate/asphalt) to partially impermeable, where intensionally made joints allow water and elements cycling. The paper is probably one of the first contributions to this topic, thus - worth of publishing. However, some important issues should be considered by authors before the final editor's decision. 1. Authors wrote about importance of joints, but readers still don't know, why they are important? Due to risk for humans or for environmental quality? Which kind of risk for humans do you mean (exposure)? This question is important if you try to apply any legal treshold! Why did you apply this for playgrounds? Is

there similar people/children exposure? Each treshold is calculated taking into account e.g. the exposure time and exposure path/way. Is the paved square comparable to any unpaved playground? 2. Authors decided to use the geochemical indexes. I'm afraid, it may not have the sense! Geochemical background, in particular in its current understanding, must be identified for soil - not for geological substratum. Background soil and soil under comparison should be comparable - also in terms of soil processes. Are the pavement joints comparable to any more or less natural soil, in terms of biological activity, bioaccumualtion processes, nutrient and water cycling? Rather not. It means, calculating the Igeo and other indexes, which were constructed taking into account real soils, has no sense. If youy cannot determine reliable geochemical background for soils under comparison - calculation of indexes whoch regire such background - is simply impossible... 3. So, any comparison to legal tresholds/intervention values and indexes have a sense if you can combine it with a kind of risk. If you cannot explain how the accumulated metals may influence humans or environment - you don't know if the scales are applicable... 4. Authors tried to combine the soil contamination in the joints with water cycling. But we know, that the pavement materials are commonly laid on the stabilised ground, often with admixture of cement, or mechanically compacted. All these stabilisations lead to impermeability. Thus, even if the pavement is not continuous, the underlying layers may be impermeable and thus all the cover is impermeable for water and roots. Such cover may have some capacity for rain/melting water (in joints and subsequent layers), but it may not mean permeability and cycling... Other authors suggest protective role of pavement for underlying soil - already due to pavement impermeablity for water and solutes... Charzyński, P., Plak, A., & Hanaka, A. (2017). Influence of the soil sealing on the geoaccumulation index of heavy metals and various pollution factors. Environmental Science and Pollution Research, 24(5), 4801-4811. Mendyk, Ł., & Charzyński, P. (2016). Soil sealing degree as factor influencing urban soil contamination with polycyclic aromatic hydrocarbons (PAHs). Soil Science Annual, 67(1), 17-23. (I do not agree with all statements and conclusions presented in the above cited papers, but I think Authors should at least read these

opinions) 5. Authors don't have informations about the mobility of metals in the joints, thus any conclusions referring the their translocation should take into account the general knowledge and confirmed affinity of (some) metals to organic matter, in particular under neutral/alkaline reaction.

СЗ

---

## Referee Comment (RC2) · Anonymous Referee #2 · 14 Sep 2020

General comments Evaluation of the overall quality of the preprint

This manuscript focuses on the accumulation of various elements, including potentially toxic metals and one metalloid, in the joints of pavements in selected sites spread in Marburg city. The problem has not been in fact dealt with or described in soil science literature, and therefore the related data are - in my opinion - worth publishing. However, the presentation of such data should be accompanied by a thorough analysis and discussion. Unfortunately, the issue has been treated in this paper quite superficially, despite the fact that the text can make the impression of a deep scientific interpretation. This is, however, only the impression. After thorough reading the manuscript, I can see a lot of its shortcomings and errors. Moreover, the overall interpretation is in my opinion improper. The first issue is whether these joints can be called soil, which

has already been raised by the editor. I myself have also doubts if it is soil, but the presented reasoning and conclusions can be considered a rational voice in the discussion. Of course, one could continue this discussion, mainly because the Technosols should generally show continuity with the rocks / parent rocks occurring underneath, while the pavements, especially the modern ones, are often placed on the completely impermeable layers equipped with drainage. However, if defining the soil as a living layer on the earth's surface, we can consider the joints a discontinuous soil layer. The authors should therefore first raise the issue whether the pavement joints make a soil or not, and then possibly move on to their position in soil classification. Regardless of whether it is soil or a questionable soil, I think the problem of accumulation of pollutants in the urban environment and a related risk assessment is worth tackling. In this manuscript, an added value is the data on concentrations of potentially toxic elements in the joints of pavements. However, the entire interpretation of the results seems to me wrong. Therefore, the results could be published after their re-interpretation. The main concerns are as follows: Firstly: the authors lump together various elements, including those that are the natural macro-components of rocks / soils / building materials: Fe or Al, and those that can be highly toxic, like Cd. The presence of As and Fe in soils is in fact of no importance in terms of pollution or risk. Moreover, their pseudototal concentrations (determined after the digestion in aqua-regia) are difficult to interpret. Secondly: the values of Igeo have been calculated in the manuscript with a purely technical procedure that should not be applied in this case. I am afraid, it would not be possible to calculate the values of Igeo index for the pavement joints. This index, proposed by Muller (1969) for river sediments, should be very carefully applied to soils. This matter was emphasized by Cai et al. in a quoted article (2015). The Igeo index has a geological meaning, and it can be applied to technically undisturbed, untransformed soils. It only makes sense to use it when there is a continuum: soil parent rock - soil. This index illustrates "geoaccumulation", which should be understood as the enrichment of geological material in comparison with its original state. In the case of pavement joints, their original "parent" material is unknown (e.g. cement, concrete,

pure sand, gravely loams etc.) and most likely it differs among various sites. The authors proposed a very strange procedure to invent a "common reference material" that deal as a "local geochemical background". I do not think it can be justified to average the background values for soils from volcanogenic substrates (n = 94) and external sands (n = 64). Furthermore, the numbers of samples taken to this calculation (n = 94, 64) have not been explained. In my opinion, it also does not make sense to calculate for pavement joints such operationally defined parameters as the pollution load index (PLI) and potential ecological risk (RI), proposed by Hakanson. The authors maintain that "All three indices allow an effective assessment of contamination and the spatial differences (Kowalska et al., 2018; Cai et al., 2015). However, in the cited paper, Cai et al. highlighted various disadvantages of those simplified indices when they are applied to soils. Many authors have applied them indiscriminately to soils, indeed, and such works have been published, but this is not an argument that can justify such a controversial approach. The use of any of those indices should be preceded by their critical analysis and the study of applicability for a given case. While they can be sometimes used for the preliminary assessment of environmental risk, they cannot be used automatically and uncritically. The part of the manuscript that focuses on the PLI and RI indices is written incomprehensibly and it is hard to follow. The authors should try to be more concise. The methods of calculation and related input data have not been clearly presented. The corresponding threshold values have not been clearly shown either. The applicability of the indicators and their substantive sense have not been discussed at all. Based on the calculated values of the above indices, the authors drew conclusions regarding the risk assessment. However, they did not define the types of risk related to the accumulation of PTEs in the joints. Such an analysis should be a real basis for any interpretation of the results. The indices of ecological risk, proposed 40 years ago by Hakanson, were applied originally to the aquatic environment. They can be sometimes used as a simplified tool for very preliminary assessment of the risk associated with pollution of terrestrial environment, though a related kind of risk should be well specified. The risk analysis should take into account the potential

human exposure pathways and the elements of possible risk to ecosystems (contaminated particles blown by wind, leaching of contaminants to groundwater, runoff and pollution of natural waters, other pathways of spread in the environment). Additionally, a surface area of pavements joints, exposure time of the target groups, potential infiltration rates (if any?), and the solubility of potentially toxic elements in joints should be taken into account and discussed to make the assessment of risk realistic. In relation to the last point, the importance of pH for the risk assessment should be clearly explained in the paper. The reasoning leads to conflicting conclusions regarding this problem. On Both acidification and alkaline pH have been reported as unfavorable factors. Summing up the overall assessment of the manuscript and my recommendations, I would suggest to narrow the interpretation of analytical data to what is in fact in the title, i.e. the concentrations of selected potentially toxic elements in the pavement joints. I would definitely remove the parts concerning Igeo and risk indices, as they are essentially unfounded and incorrect. Furthermore, I would not emphasize the analysis of spatial variability, and concentrate rather on initial recognition of the issue. Admittedly, the paper contains some elements of spatial analysis, though its statistical correctness and soundness raise my further doubts. I cannot see in which way this manuscript "improves the understanding of the spatial variability of heavy metal contaminations in pavement joints, which is necessary for the development of targeted urban land management strategies" (L. 97-98). In my opinion this statement is too general and the manuscript does not contain clear indications for further usage of presented results. Specific comments Comments to individual scientific questions/issues My particular concerns are those related to the estimation of analytical uncertainty of experimental results (analytical errors). Although there is information that the results were obtained on the basis of triplicates (L. 164), it seems to relate only to the ICP-MS analysis of digests, and not to the differentiation of metal concentrations in "soil" within a given point. The low RSD values (L. 231-2) reported in the paper characterize the precision of instrument and not the heterogeneity of "soils" within a particular joint. Moreover, there is nothing in the manuscript about validation of analytical methods (usage of certified reference materials?) Further, with regard to analytical uncertainty: it is completely unjustified and incorrect to present the concentrations of elements in soils with ridiculously high precision. Some data listed in the table 2 have 6 or 7 digits (Are they considered significant?). The number of digits should be reduced based on the assessment of uncertainty, i.e. either on the basis of minimum 3 separate soil replicates that characterize a given point, or on the basis of other available data on intrinsic heterogeneity of the tested material. Unfortunately, such an assessment may be difficult, because (as declared by the authors themselves) there are no available results of previous research on the pavement joints. On the other hand, however, the aspect of heterogeneity alone could be quite interesting. It would be advisable to list the related legal threshold values, that have been referred to, either in the manuscript itself or in the supplementary materials. I cannot find any comprehensive list of thresholds. Single values were given in the caption to Fig. 3, however they seem to be chaotically collected. I would prefer having the completed reference values in a clearer form (for instance in a separate table). Presumably, these values have been taken from the quoted BbodSchV act (Bundesregierung: 1998). It would be nice if the authors could confirm that these threshold values have not changed since 1998 and that they are currently valid. The article mentions "preventative" values for various forms of land use (L. 236-243), but the related data not been reported. Maybe this information is available in references, but the text cannot be understood without clear defining of terms used. Similarly: a term "absolute concentrations of heavy metals" appears several times in discussion (eg. L. 324). This term is unclear; it should be defined. According to the text (L. 86-88), "pollution retention capability of pavement joints has mostly been determined in laboratory tests and not in the field". This statement was based on two bibliographic sources. The related data can by no means be generalized. This issue should be discussed in more depth, as the retention capability of pavement joints can undoubtedly be very diverse, and it would govern the environmental effects caused by enrichment in metals (surface runoff vs. infiltration). This aspect seems important and should be taken into account more carefully when interpreting the data. What did the

authors understand as urban soil stratigraphy (L. 144)? Stratigraphy deals with natural rock layers (strata) and their layering (stratification), mainly in the studies of natural sedimentary and other layered rocks. I do not think this term is applicable to human made materials. The calculations of Igeo do not make sense as it is impossible to apply "the respective background value" (L. 178) The calculation of PLI was poorly explained. PLI was calculated as the square root of all multiplied single pollution indexes (L. 178), but no clear explanation was given what these single pollution indices are. The terms related to various elements should be used precisely. For example, As is not a metal but a metalloid (L. 159) and Al is not a heavy metal (L. 232). In the discussion, the authors introduced (based on their results?) the division into mobile and immobile elements (L. 395-6). This division is inconsistent with the generally known susceptibility of contaminants to mobilization in soils. Pb is known as a metal with very strong adsorption affinity and low mobility in soils. Such a grouping should be confronted with literature and discussed. One of the final conclusions, implying that basaltic rock material should be avoided as the material used for construction of pavements because of its alkaline pH and capacity to accumulate heavy metals, seems completely unjustified, particularly without a comprehensive discussion on risk exposure pathways. A list of references contains 13 items (23%). in German. I would suggest removing some of them and replacing them by English literature. If an aspect of risk assessment is to be included in the work, the literature should be supplemented with related sources

A list of technical corrections, typing errors, etc. L. 42, 83 singular "concentration" should be replaced by plural "concentrations". L. 66 – Reference: Burghardt 1995 cannot be used as the reference for 2015 WRB-FAO classification L. 80-84 Unnecessary (obvious) references: (Sorme and Lagerkvist, 2002; Sansalone et al., 1998), (Gilbert and Clausen, 2006; Wessolek et al., 2009). L. 86 (Dierkes et al., 1999; Dierkes et al., 2004; Dierkes et al., 2005a) can be simplified: (Dierkes et al., 1999; 2004; 2005). L. 91 a phrase "contain an accumulation of heavy metals: should be reworded L.157 - "Ph value" should be corrected. I would suggest replacement by "The pH value" L. 159 – "pseudo-total concentration .... was" should be replaced by "pseudo-total concentrations .... were" L. 160 A term "extraction" with aqua regia should rather be replaced by "digestion". Moreover, if a mass of soil sample is recorded (1g), a related volume of aqua regia should also be given. Or both data may be omitted. L. 161 each sample. It should be specified: each soil sample or each digest sample? The heterogeneity of soils is much larger than that of liquid samples. Definitely, solid samples should be analysed in triplicate. L. 192-3 rSP – it can be guessed that this abbreviation stands for Spearman correlation coefficient, but it should be clearly explained. Section 3.1 . (L. 195-210) Repeated discussion on soil classification is unnecessary here and should be removed. This section should focus on concentrations of elements in "soils" as declared in the title of the paper The list of references should be improved. Presently, it is drawn up inconsistently. Journal titles have been sometimes given in full names, sometimes in abbreviations, capital letters should be used. Is there any difference between Dierkes et al. 2005 a and b? The description shows that it is the same source Figure 1 – test sites are shown in Legend but not in Figures (except for the Marburg in the map of Germany) Figure 2 should be, in my opinion, divided into 2 separate graphs as they present different issues. A graph a (Igeo) presents the data concerning various elements, while two other graphs (b, c) – refer to various sites. Red lines placed in the graphs should consistently have the same meaning within one figure. Figure 3 – the graph is unclear and difficult to follow. The comprehensive data of preventive thresholds should rather be described in a table and not in the Figure description.

---

## Author Comment (AC1) · 8 Oct 2020

Dear Referee #1,

Many thanks for your time and efforts to read and comment on our current manuscript. You have raised some very important points, for which we are very grateful. According your remarks as well as the notes of Referee 2, we have changed and hopefully improved several parts of our manuscript. Below we will reply to each of your comments point by point.

1. Authors wrote about importance of joints, but readers still don't know, why they are important? Due to risk for humans or for environmental quality? Which kind of risk for humans do you mean (exposure)? This question is important if you try to apply any

legal treshold! Why did you apply this for playgrounds? Is there similar people/children exposure? Each treshold is calculated taking into account e.g. the exposure time and exposure path/way. Is the paved square comparable to any unpaved playground?

⇒ Thank you for this comment and your questions. First of all, we think that the joints are important for environmental quality in urban areas as we stated clearly for example in l. 69-75 (infiltration or partly soil functions) or in l. 226 (function of "topsoil" in sealed areas). Secondly, we think, that the potential accumulation of heavy metals found in pavement joints poses different risks to humans and the wider environment in the surrounding of urban areas. As we see your concerns about the application of different legal thresholds and the overall importance of the risk assessment, we have made different changes in our manuscript, to overcome your concerns regarding this point. First, we added an additional chapter before the conclusion, where we discuss different risks and potential exposure pathways. Out from this new risk assessment, we also discuss the usefulness of the available legal thresholds for the special case of urban pavement joints.

2. Authors decided to use the geochemical indexes. I'm afraid, it may not have the sense! Geochemical background, in particular in its current understanding, must be identified for soil - not for geological substratum. Background soil and soil under comparison should be comparable - also in terms of soil processes. Are the pavement joints comparable to any more or less natural soil, in terms of biological activity, bioaccumualtion processes, nutrient and water cycling? Rather not. It means, calculating the Igeo and other indexes, which were constrcucted taking into account real soils, has no sense. If youy cannot determine reliable geochemical background for soils under comparison - calculation of indexes whoch reqire such background – is simply impossible...

⇒ Thanks that you state this very important point. In our first version of the manuscript, we searched for a successful way how we can overcome the several problems regarding the evaluation of heavy metal loads in urban pavement joints without any comparative values. Therefore, we decided to apply some of the well-known geochemical indices with the best available background soil, knowing the many limitations of this approach. ⇒ After your concerns and the concerns of Referee 2, which stated the same point, we decided to remove the calculation of Igeo, PLI and RI completely! Instead we are working with the absolute concentrations (given in mg/kg) within our assessment and interpretation. For the analyses of spatial relationships, we decided to calculate the ExF according to BĂĚbelewska (2010), as it provides an information where, in a given study area, the highest metal loads are located. We think, that the calculation of this index is appropriate, as it is based on absolute metal concentrations and average contents at each sampling site (without any geochemical background value). Calculation and reasons for the selection of this index will be stated clearly in the revised method section.

3. So, any comparison to legal tresholds/intervention values and indexes have a sense if you can combine it with a kind of risk. If you cannot explain how the accumulated metals may influence humans or environment - you don't know if the scales are applicable...

⇒ Thank you for this remark. In general, we think that the accumulated heavy metals could pose a risk to humans on different pathways (e.g., direct soil-human contact in the case of playing children or the indirect contact including soil-air pathway by dust emissions). We stated this point in l. 74 or l. 419. Regarding the environmental risks, we see the main risk in the accumulation and therefore storage of heavy metals in pavement joints and a potential output through surface runoff (e.g., during stormwater events). In l. 355-361 we clarified, that urban surface runoff seems to play an important role for the spatial distribution of heavy metals in pavement joints. Therefore, it is thinkable, considering the existing literature as well (e.g., Drake et al., 2014 or Wessolek et al. 2011), that accumulated heavy metals could be relocated during stormwater events and reach urban surrounding areas (like river systems, floodplains). This point was stated in l. 417-418. ⇒ Since we noticed that this point was apparently not communicated clearly enough, we made it clear at various points in the manuscript what kind of risk is meant. In addition, as mentioned above, we have included the additional chapter on risk assessment.

4. Authors tried to combine the soil contamination in the joints with water cycling. But we know, that the pavement materials are commonly laid on the stabilised ground, often with admixture of cement, or mechanically compacted. All these stabilisations lead to impermeability. Thus, even if the pavement is not continuous, the underlying layers may be impermeable and thus all the cover is impermeable for water and roots. Such cover may have some capacity for rain/melting water (in joints and subsequent layers), but it may not mean permeability and cycling... Other authors suggest protective role of pavement for underlying soil - already due to pavement impermeablity for water and solutes... Charzy′ nski, P., Plak, A., & Hanaka, A. (2017). Influence of the soil sealing on the geoaccumulation index of heavy metals and various pollution factors. Environmental Science and Pollution Research, 24(5), 4801-4811. Mendyk, Ł., & Charzy′ nski, P. (2016). Soil sealing degree as factor influencing urban soil contamination with polycyclic aromatic hydrocarbons (PAHs). Soil Science Annual, 67(1), 17-23. (I do not agree with all statements and conclusions presented in the above cited papers, but I think Authors should at least read these opinions)

⇒ Many thanks for this comment and especially for the literature references. We enjoyed reading the papers and reflecting on the opinions. First of all, a small clarification seems to be necessary at this point: If we are talking about "surface runoff" (line 336) or "urban drainage and surface runoff with stormwater runoff" (line 353) as a potential transport medium of heavy metals and a possible factor that explains the enrichment at certain points (e.g., lowest points, drainage accumulation points), then permeability through underlaying layers plays only a minor role. Of course, you are right, that pavement materials laid on a stabilized ground. In the case of our study area, we found crushed stone (coarse and fine stones) as well as sand under the pavement, without cement (line 220). These materials are stabilized, but not impermeable for water and

roots. In general, the underlaying layers have to be separated from the joints itself, as the pavement joints show totally different characteristics like organic material and more heterogeneous grain sizes, which enables capacity for rain water / surface runoff as well as heavy metal retention (line 404). ⇒ Regarding the "water cycling" we think that a partwise infiltration of water, possibly with dissolved heavy metals, in underlaying layers, could be possible, as one of main function of pavement areas is the partly permeability compared to fully sealed surfaces (line 436). More important for us, however, is the conclusion that heavy metals that accumulate in the pavement joint material can be washed out again directly on the surface by heavy rain events. An infiltration to underlaying soils is not needed, as the surface runoff in urban areas could be directly reach the sewerage at the pavement surface (line 418). If this runoff has absorbed heavy metals from pavement joints, the contamination can reach other places inside the urban area or influence river ecosystems in urban surroundings. ⇒ The "protective role" of pavement for underlying soils, stated in the publication of Charzy′ nski et al. (2017) is an interesting opinion, but not directly transferable to our study area, as the side constructions seems to be very different. Again, in our recent study, we didn't try to combine heavy metals loads in pavement joints with insurance to deeper soil layers by water. We have considered only processes that occur at the surface or pavement joint layer and that can be important for a) the spatial distribution of the metal concentrations or b) the discharge out of urban areas (over a large area).

5. Authors don't have informations about the mobility of metals in the joints, thus any conclusions referring the their translocation should take into account the general knowledge and confirmed affinity of (some) metals to organic matter, in particular under neutral/alkaline reaction.

⇒ Finally, we would like to thank you also for this last remark. Of course, we don't have information about the mobility of meals in joints. In chapter 3.4 of our manuscript (line 367) we discuss the possible sources and translocation tendencies of different metals out from a correlation with organic matter and pH milieu. The discussion as well as the

conclusions out of this paragraph are based, exactly as you demand in your comment, on the general knowledge about heavy metal mobility in soils.

Sincerely,

Collin J. Weber (on behalf of the authors)

Please also note the supplement to this comment:
https://soil.copernicus.org/preprints/soil-2020-39/soil-2020-39-AC1-supplement.pdf

---

## Author Comment (AC2) · 8 Oct 2020

Dear Referee 2, special thanks for your in-depth review on our current manuscript. As this extensive review has certainly taken a lot of time, we would like to thank you especially for this. We greatly appreciate your detailed comments and remarks. They have led us to rethink some aspects and approach several parts of the manuscript in a completely new way. As we are working on the revised manuscript in parallel to the review process, we have attached some new or revised figures and tables to our answer (you will find this figures at the end of the supplement .pdf-file). In the following we will respond to your individual comments, one by one.

1. The first issue is whether these joints can be called soil, which has already been

raised by the editor. I myself have also doubts if it is soil, but the presented reasoning and conclusions can be considered a rational voice in the discussion. Of course, one could continue this discussion, mainly because the Technosols should generally show continuity with the rocks / parent rocks occurring underneath, while the pavements, especially the modern ones, are often placed on the completely impermeable layers equipped with drainage. However, if defining the soil as a living layer on the earth's surface, we can consider the joints a discontinuous soil layer. The authors should therefore first raise the issue whether the pavement joints make a soil or not, and then possibly move on to their position in soil classification. → Thanks for this first comment regarding the "soil definition question". We will improve the discussion about this important question in our manuscript, staring with a general discussion if those material could be classified as soil (considering soil functions) and then follow your advice to move on our own position regarding this question.

2. Regardless of whether it is soil or a questionable soil, I think the problem of accumulation of pollutants in the urban environment and a related risk assessment is worth tackling. In this manuscript, an added value is the data on concentrations of potentially toxic elements in the joints of pavements. However, the entire interpretation of the results seems to me wrong. Therefore, the results could be published after their re-interpretation. The main concerns are as follows: → Thank you very much for this important remark, including the following sub-comments. In general, we decided to examine and reconsider our interpretation. Suggested changes will be stated in answers to the next sub-comments.

2.1 Firstly: the authors lump together various elements, including those that are the natural macro-components of rocks / soils / building materials: Fe or Al, and those that can be highly toxic, like Cd. The presence of As and Fe in soils is in fact of no importance in terms of pollution or risk. Moreover, their pseudototal concentrations (determined after the digestion in aqua-regia) are difficult to interpret. → We reduced the number of elements for risk assessment and evaluation to 7, namely Cr, Ni, Cu, Sn,

[Figure]

As, Cd and Pb (see new Table . Special focus will be given to Cd, Pb, Cr, Ni and As in comparison with other values (background values of natural soils, legal values) . The selection of those 6 heavy metals and the metalloid As, is based on the elements that are included in the current German legislation and can potentially be toxic to humans or have in general negative effects on soils and the environment (of course considering the respective concentration). Additionally, the difficulties regarding pseudototal concentrations coming from the digestion in aqua-regia, is a well-known problem, which is not even mentioned by many other authors. We make this limitation clear in our method section.

2.2 Secondly: the values of Igeo have been calculated in the manuscript with a purely technical procedure that should not be applied in this case. I am afraid, it would not be possible to calculate the values of Igeo index for the pavement joints. This index, proposed by Muller (1969) for river sediments, should be very carefully applied to soils. This matter was emphasized by Cai et al. in a quoted article (2015). The Igeo index has a geological meaning, and it can be applied to technically undisturbed, untransformed soils. It only makes sense to use it when there is a continuum: soil parent rock - soil. This index illustrates "geoaccumulation", which should be understood as the enrichment of geological material in comparison with its original state. In the case of pavement joints, their original "parent" material is unknown (e.g. cement, concrete, pure sand, gravely loams etc.) and most likely it differs among various sites. The authors proposed a very strange procedure to invent a "common reference material" that deal as a "local geochemical background". I do not think it can be justified to average the background values for soils from volcanogenic substrates (n = 94) and external sands (n = 64). Furthermore, the numbers of samples taken to this calculation (n = 94, 64) have not been explained. In my opinion, it also does not make sense to calculate for pavement joints such operationally defined parameters as the pollution load index (PLI) and potential ecological risk (RI), proposed by Hakanson. The authors maintain that "All three indices allow an effective assessment of contamination and the spatial differences (Kowalska et al., 2018; Cai et al., 2015). However, in the cited paper,

[Figure]

Cai et al. highlighted various disadvantages of those simplified indices when they are ap- plied to soils. Many authors have applied them indiscriminately to soils, indeed, and such works have been published, but this is not an argument that can justify such a controversial approach. The use of any of those indices should be preceded by their critical analysis and the study of applicability for a given case. While they can be some- times used for the preliminary assessment of environmental risk, they cannot be used automatically and uncritically.

$\rightarrow$ As our manuscript is one of the first trials dealing with the topic of metal accu- mulation in urban pavement joints, we tried to perform a data preparation with some well-known indices in the field of soil science. In general, as a first attempt. Regarding your concerns as well as the concerns of Referee 1, we decided to cancel the calcu- lation of Igeo, as well as PLI and RI. Some of the concerns you have presented, we have had ourselves before and have tried to overcome them as best we can (with no preliminary studies or comparative data). We are also well aware of the fundamental limitations of so called "simple" indices. Instead of the usage of the above-mentioned indices, we decided to work with the absolute values (given in mg/kg) for the selection of 7 elements. For the comparison between absolute terrain heights (a.s.l ) and metal contents (former figure 4) we decided to make a new calculation instead of the PLI. We calculated the Exposure factor (ExF) according to BÄĚbelewska (2010), as it provides an information where, in a given study area, the highest metal loads are located. We think, that the calculation of this index is appropriate, as it is based on absolute metal concentrations and average contents at each sampling site (without any geochemical background value). Calculation and reasons for the selection of this index will be stated clearly in the revised method section.

2.3 The part of the manuscript that focuses on the PLI and RI indices is written incom- prehensibly and it is hard to follow. The authors should try to be more concise. The methods of calculation and related input data have not been clearly presented. The corresponding threshold values have not been clearly shown either. The applicability

of the indicators and their substantive sense have not been discussed at all. → The section regarding the PLI and RI indices was rededicated, as we omit the indices (as mentioned above). Instead we restructured this section in a more concise way, as you requested. The discussion on the spatial variability of heavy metals in our study area is now based on the absolute heavy metal concentrations (for correlation analyses) and the Exposure factor for the linear regression with absolute terrain heights. Calculation of ExF (Figure 4), data for distances to next road (Tab. 3) and calculation of the runoff accumulation value (Tab. 4) is now clearly presented in the method section. Additional thanks for this important remark!

3. Based on the calculated values of the above indices, the authors drew conclusions regarding the risk assessment. However, they did not define the types of risk related to the accumulation of PTEs in the joints. Such an analysis should be a real basis for any interpretation of the results. The indices of ecological risk, proposed 40 years ago by Hakanson, were applied originally to the aquatic environment. They can be sometimes used as a simplified tool for very preliminary assessment of the risk associated with pollution of terrestrial environment, though a related kind of risk should be well specified. → Thanks for this comment. We recognize that we must pay more attention to risk assessment. Therefore, we have decided, after previous sections have been condensed, to include a separate chapter on risk assessment before the conclusion. In this section we will clearly define the types of risk related to the heavy metal accumulation in pavement joints and will discuss the known pathways of risks one by one. Additionally, we have created a new figure that illustrates the risk on various impact and human interaction paths (see new Figure 6).

3.1 The risk analysis should take into account the potential human exposure pathways and the elements of possible risk to ecosystems (contaminated particles blown by wind, leaching of contaminants to groundwater, runoff and pollution of natural waters, other pathways of spread in the environment). → As mentioned above, we included the potential pathways for human exposure and environmental risks in the new chapter

and Figure 6.

3.2 Additionally, a surface area of pavements joints, exposure time of the target groups, potential infiltration rates (if any?), and the solubility of potentially toxic elements in joints should be taken into account and discussed to make the assessment of risk realistic. → Very important note! Thank you. We included the surface area, the potential exposure time and the solubility of potential toxic elements in our discussion. Infiltration rates are not available for our locations. However, we were able to draw on some other work as a comparison to infiltration on paved areas (e.g. Bean et al., 2007 doi/10.1061/(ASCE)0733-9437(2007)133%3A3(249) or Chopra et al., 2010 10.1061/(ASCE)HE.1943-5584.0000117 ).

3.3 In relation to the last point, the importance of pH for the risk assessment should be clearly explained in the paper. The reasoning leads to conflicting conclusions regarding this problem. On Both acidification and alkaline pH have been reported as unfavorable factors. → The effect of the overall neutral to slight alkaline pH in the studied pavement joints was discussed in section 3.4 "Accumulation and mobility tendencies of heavy metals in pavement joints". We have already stated that pH conditions support a fixation of metals by specific bindings in contrast to acid conditions. We have reworded this passage to make the statements clearer.

4. Summing up the overall assessment of the manuscript and my recommendations, I would suggest to narrow the interpretation of analytical data to what is in fact in the title, i.e. the concentrations of selected potentially toxic elements in the pavement joints. I would definitely remove the parts concerning Igeo and risk indices, as they are essentially unfounded and incorrect. Furthermore, I would not emphasize the analysis of spatial variability, and concentrate rather on initial recognition of the issue. → Thank you for summarizing the important remarks and your suggestion. As stated in our comments above, we are working on the shortening and rewriting of the manuscript based on your suggestions as well as the comments from Referee 1. Igeo and risk indices are removed, new figures and tables have already been prepared. Regarding the spatial
variability, we would like to point out that precisely this point is of particular importance to us as soil geographers. The determination and risk assessment for pavement joints alone does not seem sufficient for us, unless we also address spatial issues. These are particularly important for possible leaching, but also for risk management. Nevertheless, we think that with the help of your comments this discussion and interpretation will also be significantly improved.

5. Admittedly, the paper contains some elements of spatial analysis, though its statistical correctness and soundness raise my further doubts. I cannot see in which way this manuscript "improves the understanding of the spatial variability of heavy metal contaminations in pavement joints, which is necessary for the development of targeted urban land management strategies" (L. 97-98). In my opinion this statement is too general and the manuscript does not contain clear indications for further usage of presented results. → Continue with the comment above: Besides a basic pollution assessment, the second goal of our study was to clarify possible sources and mobilization on the basis of this data set, based on a geospatial approach (see for example: Weihrauch 2019 https://doi.org/10.1016/j.geoderma.2019.05.025 or Weber et al. 2020 https://doi.org/10.1002/ldr.3676 ). Already in the selection of the individual study sites, possible heavy metal sources and different exposures to them were taken into account. The sampling sites were also selected in such a way that a possible influence of road emissions or transport through surface runoff can be considered. We think that a discussion and interpretation of spatial distribution patterns is possible on the basis of our data, with the given limitations. Even though the sample is comparatively small, we have been able to demonstrate clear and correct statistical correlations that allow a first interpretation of the spatial patterns found. However, as we understand your concerns, we will rephrase the sections and make our statements more carefully and with more reference to our investigation area.

Specific comments comments to individual scientific questions/issues

6. My particular concerns are those related to the estimation of analytical uncertainty

of experimental results (analytical errors). Although there is information that the results were obtained on the basis of triplicates (L. 164), it seems to relate only to the ICP-MS analysis of digests, and not to the differentiation of metal concentrations in "soil" within a given point. The low RSD values (L. 231-2) reported in the paper characterize the precision of instrument and not the heterogeneity of "soils" within a particular joint. Moreover, there is nothing in the manuscript about validation of analytical methods (us- age of certified reference materials?)  → Thanks for this comment. Of course, you are right that the method section regarding the ICP-MS analyses is to short and raises concerns. It is correct that these are triplicates, related to the analyses of digest. We have clarified this point in the section. The low RSD values mentioned, were not interpreted by us as heterogeneity of the "soil", but were used as precision control of the individual measurements. We have also rewritten this point. In general, we added a paragraph about the used certified reference materials, standards for metal analysis and standard concentrations to improve the validation of analytical methods. A final assessment of uncertainty was added, based on the available data and controls.

6.1 Further, with regard to analytical uncertainty: it is completely unjustified and incorrect to present the concentrations of elements in soils with ridiculously high precision. Some data listed in the table 2 have 6 or 7 digits (Are they considered significant?). The number of digits should be reduced based on the assessment of uncertainty, i.e. either on the basis of minimum 3 separate soil replicates that characterize a given point, or on the basis of other available data on intrinsic heterogeneity of the tested material. → We apologies for this mistake. Digits were reduced to 1 for mean, median, min. or max. values calculated from the dataset.

6.2 Unfortunately, such an assessment may be difficult, because (as declared by the authors themselves) there are no available results of previous research on the pavement joints. On the other hand, however, the aspect of heterogeneity alone could be quite interesting. It would be advisable to list the related legal threshold values, that have been referred to, either in the manuscript itself or in the supplementary materials.

I cannot find any comprehensive list of thresholds. Single values were given in the caption to Fig. 3, however they seem to be chaotically collected. I would prefer having the completed reference values in a clearer form (for instance in a separate table). Presumably, these values have been taken from the quoted BbodSchV act (Bundesregierung: 1998). It would be nice if the authors could confirm that these threshold values have not changed since 1998 and that they are currently valid. → Thank you very much for this suggestion. As you can see in the new Table 2 as well as in Figure 2, we have now added the related legal threshold values and geochemical background values for natural soils in a clear way. First, the values taken from BBodSchV 1998 are currently valid and the only legal threshold values for heavy metal concentrations in soil in Germany. Second, as we should compare the concentrations on a more fundamental level, we have decided to use the following values for comparison: 1) Average content surface horizons worldwide according to Kabata-Pendias (2011), 2) Composition in upper continental crust according to Rudnick & Gao (2003), 3) Geochemical background in Hessian soils with an calculation explained in method section according Friedrich & Lügger (2011) as well as 4) the legal precautionary level for residential areas according to the German Federal Soil Protection Ordinance - BBodSchV (1998).

6.3 The article mentions "preventative" values for various forms of land use (L. 236-243), but the related data not been reported. Maybe this information is available in references, but the text cannot be understood without clear defining of terms used. → The German Federal Soil Protection Ordinance - BBodSchV (1998) differentiates in the threshold values according to land use and thus potential exposure to humans. To avoid further discrepancies, we are now considering only the legal precautionary level for residential areas and the preventative value for sandy substrates (not differentiated to land use forms by the law). Explanations can now be found in the method section.

6.4 Similarly: a term "absolute concentrations of heavy metals" appears several times in discussion (eg. L. 324). This term is unclear; it should be defined. → Thank you for this remark. The term was used, to clarify the distinction between measured concentrations (given in mg/kg) and calculated index values. As the index values (Igeo, PLI, RI) are now removed, we have changed the terms to "concentrations of heavy metals".

6.5 According to the text (L. 86-88), "pollution retention capability of pavement joints has mostly been determined in laboratory tests and not in the field". This statement was based on two bibliographic sources. The related data can by no means be generalized. This issue should be discussed in more depth, as the retention capability of pavement joints can undoubtedly be very diverse, and it would govern the environmental effects caused by enrichment in metals (surface runoff vs. infiltration). This aspect seems important and should be taken into account more carefully when interpreting the data. → Thanks again for this important note. We will try to discuss this point more carefully and take it into account for our further interpretation, especially in the risk assessment and estimation of potential exposure pathways. We have changed the sentence to avoid that the statement becomes too general.

6.6. What did the authors understand as urban soil stratigraphy (L. 144)? Stratigraphy deals with natural rock layers (strata) and their layering (stratification), mainly in the studies of natural sedimentary and other layered rocks. I do not think this term is applicable to human made materials. → On this point we must unfortunately disagree. The term "urban soil stratigraphy" or now rephrased "stratigraphy of urban soils" means the description of stratification or layering of pavement joint material. The material, even if anthropogenic deposited, has its own stratification which should be documented, the same as for natural soils. We think it is permissible to transfer existing concepts to an anthropogenic material if the properties can be compared and only the deposition process is not natural (e.g., Zalasiewicz et al. 2018, https://doi.org/10.1016/j.pgeola.2017.06.004 ).

6.7 The calculations of Igeo do not make sense as it is impossible to apply "the respective background value" (L. 178) The calculation of PLI was poorly explained. PLI was calculated as the square root of all multiplied single pollution indexes (L. 178), but no clear explanation was given what these single pollution indices are. The terms related

to various elements should be used precisely. For example, As is not a metal but a metalloid (L. 159) and Al is not a heavy metal (L. 232). → Thank you for drawing our attention to the terms related to the focused elements. We have carefully checked each wording inside the text. Igeo and PLI are removed, as stated above.

6.8 In the discussion, the authors introduced (based on their results?) the division into mobile and immobile elements (L. 395-6). This division is inconsistent with the generally known susceptibility of contaminants to mobilization in soils. Pb is known as a metal with very strong ad- sorption affinity and low mobility in soils. Such a grouping should be confronted with literature and discussed. One of the final conclusions, implying that basaltic rock material should be avoided as the material used for construction of pavements because of its alkaline pH and capacity to accumulate heavy metals, seems completely unjustified, particularly without a comprehensive discussion on risk exposure pathways. → The division into mobile and immobile elements (l. 395-6.) is based on the inter-element relationships resulting from our findings (Fig. 5, Spearman correlation matrix) and also performed by Manta et al. (2002) for example. We see your concerns and have therefore expanded this section to include further literature and a discussion of this result. Including the new chapter containing the discussion of risk exposure pathways, we will review the conclusion regarding the avoidance of basaltic rock material.

7. A list of references contains 13 items (23%). in German. I would suggest removing some of them and replacing them by English literature. If an aspect of risk assessment is to be included in the work, the literature should be supplemented with related sources → Unfortunately, the "high" percentage of references in German is due to the fact that references regarding the study area description (4) as well as references regarding the legislation (laws, regulations, background values and standards) (5) are not available in English. Other references will be replaced by English literature. Additional references regarding the risk assessment will be added.

A list of technical corrections, typing errors, etc. → Thank you for the list of technical

or typing errors, as well as your recommendations! We have implemented the remarks in the manuscript.

L. 42, 83 singular "concentration" should be replaced by plural "concentrations". → The correction has been adopted in the respective lines. L. 66 – Reference: Burghardt 1995 can- not be used as the reference for 2015 WRB-FAO classification → The reference Burghardt 1995 was removed. L. 80-84 Unnecessary (obvious) references: (Sorme and Lagerkvist, 2002; Sansalone et al., 1998), (Gilbert and Clausen, 2006; Wessolek et al., 2009). → The double references have been removed. L. 86 (Dierkes et al., 1999; Dierkes et al., 2004; Dierkes et al., 2005a) can be simplified: (Dierkes et al., 1999; 2004; 2005). → The respective reference has been simplified according your recommendation. L. 91 a phrase "contain an accumulation of heavy metals: should be reworded → The sentence was reworded as follows "Apart from this, the question of whether already installed and used pavement joints, not only in car parks but also, for example on pavements, contain accumulated heavy metals, still remains unclear (Dierkes et al., 2005b). L.157 - "Ph value" should be corrected. I would suggest replacement by "The pH value" → Follow your recommendation, the term Ph value was corrected to "The pH value" L. 159 – "pseudo-total concentration was" should be replaced by "pseudo-total concentrations were" → The phrase was corrected following your recommendation. L. 160 A term "extraction" with aqua regia should rather be replaced by "digestion". Moreover, if a mass of soil sample is recorded (1g), a related volume of aqua regia should also be given. Or both data may be omitted. → Thanks for this important note. We changed the term "extraction" to "aqua regia digestion" and added the related volume of aqua regia in the sentence. L. 161 each sample. It should be specified: each soil sample or each digest sample? The heterogeneity of soils is much larger than that of liquid samples. Definitely, solid samples should be analysed in triplicate. → We specified the term to "each digest of a soil sample". L. 192-3 rSP – it can be guessed that this abbreviation stands for Spearman correlation coefficient, but it should be clearly explained. → The abbreviation "rSP (Spearman correlation coefficient)" was explained after its first mention. Section 3.1 . (L. 195-210) Repeated

discussion on soil classification is unnecessary here and should be removed. This section should focus on concentrations of elements in "soils" as declared in the title of the paper. → Thank you for this comment. The section was rewritten and shorten. Now we state only the structure of pavement joint material and classification according WRB without any further discussion. The list of references should be improved. Presently, it is drawn up inconsistently. Journal titles have been sometimes given in full names, sometimes in abbreviations, capital letters should be used. → The list of references was improved according your advice. The titles are now uniformly advertised with capital letters. Is there any difference be- tween Dierkes et al. 2005 a and b? → Thank you for your accuracy! The reference Dierkes et al 2005 b was removed. The description shows that it is the same source Figure 1 – test sites are shown in Legend but not in Figures (except for the Marburg in the map of Germany) → The source "OpenStreetMap contributors 2020" is valid for all maps, except the overview map in the upper left corner. Here the corresponding source has been added. The legend was adapted, as the "test site" refers to the entire city. This term was removed in the legend. Now, traffic census stations and sampling areas are remaining. Figure 2 should be, in my opinion, divided into 2 separate graphs as they present different issues. A graph a (Igeo) presents the data concerning various elements, while two other graphs (b, c) – refer to various sites. Red lines placed in the graphs should consistently have the same meaning within one figure. → Please see the new Figure 2 as described in the comments above. In the new figure, all lines have the same meaning with the position after the corresponding values. Figure 3 – the graph is unclear and difficult to follow. The comprehensive data of preventive thresholds should rather be described in a table and not in the Figure description. → According your recommendation, we decided to reduce the figure to the upper part (a). We added a table to the supplementary material where each threshold used in Figure 3 is described. Sincerely,

Collin J. Weber (on behalf of the authors)

Please also note the supplement to this comment:

https://soil.copernicus.org/preprints/soil-2020-39/soil-2020-39-AC2-supplement.pdf

---

## Author Response (AR1)

**Response to Topical Editor Decision: Revision (19 Oct 2020)**

Dear Topical Editor,

thank you for your time and effort for reading and evaluating our manuscript. In the revised version now available (track changed version below), we have implemented all changes, as we presented in the replies to the comments of the two anonymous reviewers. In addition, we have consistently executed the change you requested regarding the name and classification of the pavement joint material. Thank you also at this point for your renewed explanations of the problem of soil definition and thank you for your understanding of our geographical background. We think that our paper has now been significantly improved and would like to thank the two reviewers and you for all your contributions.

Best regards,

Collin J. Weber
(on behalf of the authors)

[revised manuscript text omitted]